# Central Facial Nervous System Biomolecules Involved in Peripheral Facial Nerve Injury Responses and Potential Therapeutic Strategies

**DOI:** 10.3390/antiox12051036

**Published:** 2023-05-01

**Authors:** Jae-Min Lee, You Jung Choi, Myung Chul Yoo, Seung Geun Yeo

**Affiliations:** 1Department of Otorhinolaryngology, Head & Neck Surgery, College of Medicine, Kyung Hee University Medical Center, Seoul 02447, Republic of Korea; sunjaesa@hanmail.net (J.-M.L.); choiyj9825@naver.com (Y.J.C.); 2Department of Physical Medicine & Rehabilitation, College of Medicine, Kyung Hee University, Seoul 02447, Republic of Korea; famousir@naver.com

**Keywords:** peripheral facial nerve injury, facial paralysis treatment, central nervous system, biomolecules, facial motor neurons

## Abstract

Peripheral facial nerve injury leads to changes in the expression of various neuroactive substances that affect nerve cell damage, survival, growth, and regeneration. In the case of peripheral facial nerve damage, the injury directly affects the peripheral nerves and induces changes in the central nervous system (CNS) through various factors, but the substances involved in these changes in the CNS are not well understood. The objective of this review is to investigate the biomolecules involved in peripheral facial nerve damage so as to gain insight into the mechanisms and limitations of targeting the CNS after such damage and identify potential facial nerve treatment strategies. To this end, we searched PubMed using keywords and exclusion criteria and selected 29 eligible experimental studies. Our analysis summarizes basic experimental studies on changes in the CNS following peripheral facial nerve damage, focusing on biomolecules that increase or decrease in the CNS and/or those involved in the damage, and reviews various approaches for treating facial nerve injury. By establishing the biomolecules in the CNS that change after peripheral nerve damage, we can expect to identify factors that play an important role in functional recovery from facial nerve damage. Accordingly, this review could represent a significant step toward developing treatment strategies for peripheral facial palsy.

## 1. Introduction

The facial nerve, also known as the seventh cranial nerve (CN VII), is a mixed nerve composed of motor, sensory, and parasympathetic nerve fibers [1]. The facial nerve not only carries nerve impulses that control the muscles responsible for facial expression and eye blinking, but they also regulate tear and salivary gland secretion, tongue movement, and sensation of the soft palate [2]. Because the facial nerve is longer than other cranial nerves and passes through a narrow canal as it exits the temporal bone, it is more vulnerable to damage caused by middle ear and temporal bone surgery, trauma, and infection [3]. Central facial palsy, a condition in which damage to the brain by various factors, such as infection, tumor growth, or brain tumor, results in the appearance of symptoms on the contralateral side of the face and a reduction in facial muscle function [4]. In peripheral facial paralysis, functional abnormalities of the facial nerve itself appear, and symptoms occur on the same side (ipsilateral) as the lesion, resulting in reduced facial muscle function [5].

Damage to the peripheral nervous system can result in a loss of function and a reduced quality of life owing to impaired motor function [6]. One manifestation of this functional loss is peripheral facial nerve palsy, which causes paralysis of the facial expression muscles on one side [7]. About 2 in 10,000 people are diagnosed with facial nerve injury (FNI) annually, and approximately 1 in 60 will experience it during their lifetime [8]. The incidence is reported to be highest between the ages of 15 and 45, and there is no difference in frequency between men and women [9]. Facial nerve damage has different disabling symptoms depending on the site, but one of the main causes of peripheral facial palsy is neurotmesis, mostly as a result of trauma, tumor resection, or iatrogenic damage of the peripheral facial nerve [10]. Neurotmesis is the most severe type of nerve injury, in which the nerve fibers are completely severed and the surrounding connective tissue, including the epineurium, perineurium, and endoneurium, is disrupted [11]. Because this results in the loss of nerve continuity, complete regeneration is required for functional recovery [12]. After axonal transection in the peripheral nervous system, a process called Wallerian degeneration occurs that involves degeneration of the axon and myelin sheath distal to the injury site, resulting in the release of a large amount of myelin debris, which can impede axonal regeneration and functional recovery [13]. Wallerian degeneration also leads to the activation and proliferation of Schwann cells and macrophages, which phagocytose the debris, clearing the damaged tissue and providing an environment conducive to nerve regeneration [14]. Schwann cells also produce growth factors and extracellular matrix molecules that support axonal regeneration [15]. Additionally, axonal sprouts that reach the distal end of the damaged nerve can form new synapses with denervated target cells, allowing for functional recovery over time [16].

Nerve regeneration in the peripheral nervous system can lead to functional recovery, but the extent of recovery can vary depending on the severity and location of the nerve injury [6]. Although peripheral nerves have some capacity for neural plasticity and regeneration, the process is often slow and incomplete, and only partial functional recovery may be achieved [17]. Injuries that are closer to the spinal cord or involve larger nerve trunks may have a more significant impact on functional recovery because of the greater complexity and length of the nerve pathway [18]. Damaged peripheral neurons regenerate axons over a long period of time at a very slow rate (~1 mm per day). If the rate of regeneration is slow, it may take months or even years for the reinnervation of functional motor units or sensory organs to occur [19]. Following peripheral nerve injury, a number of factors involved in the process of axon regeneration show immediate up- or downregulated expression [20]. Such changes in the expression of regeneration-related genes, together with the degeneration and removal of myelin and axons, are the main features of the Wallerian degeneration process [21].

Following peripheral nerve injury, damage signals cause changes in the expression of genes encoding proteins that activate the regenerative response in neurons [22]. This leads to changes in the synthesis of molecules, including neurotransmitters, cytoskeletal proteins, and growth-associated proteins, in the central nervous system (CNS) [23]. In adult rats, transection of the facial nerve results in motor neuron damage, and after ischemic peripheral facial paralysis, the expression of genes, such as c-Jun as well as growth-associated proteins, are altered in facial nerve nuclei [24]. These alterations result in changes in various other biomolecules, including cytoskeletal molecules, metabolic enzymes, neuropeptides, and cytokines. Additionally, the expression of choline acetyltransferase, which is related to motor neuron function, is reported to be reduced in the motor nucleus of the brainstem [25].

In this review, we focus on experimental studies on changes in the CNS associated with facial nerve damage, investigating biomolecules reported to increase or decrease in the CNS following facial nerve damage in studies published between the years 2000 and 2022. Studies were identified based on the authors’ search terms and were retrieved from SCOPUS and PubMed electronic databases. The search terms used were “facial nerve damage”, “facial paralysis”, and “central nervous system”. Studies in languages other than English were excluded from the review, as were pilot studies. After applying exclusion criteria, a total of 29 eligible studies were identified. In this review, we summarize the methods and results reported by qualifying studies, highlighting changes that occur in the CNS after peripheral nerve damage, including up- and downregulation of proteins and related biomolecules, with the goal of providing insights that inform the treatment of peripheral nerve damage.

## 2. Changes in the CNS after Peripheral Facial Nerve Injury

### 2.1. Changes in Facial Motor Nerves

Peripheral facial nerve injury can cause partial or complete loss of motor, sensory, and autonomic functions in affected areas of the body, reflecting the interruption of nerve fibers and the resulting disruption in axonal continuity and degeneration [7]. After facial nerve axonal injury, the range and number of synaptic terminals of motor neurons in the facial nucleus are reduced [26]. The degeneration of motor neurons is often accompanied by the activation of microglia located in the facial nucleus [27]. The facial motor control circuitry in the CNS is able to recover to some extent after peripheral nerve damage, as evidenced by the preservation of the overall structure of the facial nucleus and the formation of reticular formations from the pons to the medulla by various types of multipolar neurons after recovery from facial nerve palsy [4]. This also implies that the neuronal circuitry responsible for facial motor control may be less vulnerable to injury and more adaptable to changes compared with other neuronal circuits in the CNS [28]. In congenital facial palsy, magnetic resonance imaging (MRI) has revealed a correlation between central and peripheral factors, and lesions have been observed in the facial nucleus of the pons [29]. Using electromyography, researchers have observed electrical signals indicative of hyperexcitability in the facial nucleus for up to 3 weeks after facial nerve compression [30]. A complete loss of neural activity, as observed following neurotmesis, is unlikely to occur during facial palsy, which is more commonly associated with axonotmesis, in which the nerve remains intact but the axons are damaged. The firing rate of motoneurons in the facial nucleus may be altered, usually reduced, compared to their original firing rate in cases of axonotmesis [10].

### 2.2. Cortical Reorganization

Neuronal plasticity is a common characteristic of the nervous system that enables neurons to adapt and modify their structure and function in response to various environmental signals, learning processes, injury, and disease [31]. Several studies have described functional plasticity in the context of various pathologies, including brain lesions and peripheral nerve transection [32,33]. This plasticity helps to restore damaged peripheral nerves by establishing an effective connection between the nervous system and the target tissue and by regulating the functional remodeling of the CNS [34]. Reconstruction in the spinal cord, brainstem, thalamus, and cortex following peripheral nerve damage has been confirmed using brain imaging techniques [35]. Cortical reorganization or cortical plasticity, which refers to the brain’s ability to change its neural connections and function in response to new experiences or changes in the environment, is a particularly evident phenomenon in the cortex—the outer layer of the brain responsible for higher cognitive functions such as perception, language, and memory [36]. Cortical plasticity plays a crucial role in learning and memory, recovery from brain injury, and adaptation to changes in sensory input [37]. Cortical reorganization has been observed in recovering Bell’s palsy patients by monitoring brain activity using functional magnetic resonance imaging (fMRI) during finger and facial movements [38]. During Bell’s palsy recovery period, there is an increase in connectivity between the ipsilateral and contralateral anterior cingulate cortex and a strengthening of the functional relationship between the unaffected anterior cingulate cortex and the sensorimotor area that contributes to adjustments in abnormal facial movements [39]. This improved connectivity of the ipsilateral and contralateral anterior cingulate cortex is the result of monitoring and compensatory functions [40]. Motor tasks performed by both paralyzed and non-paralyzed facial nerves in individuals with Bell’s palsy were found to be associated with the activation of several brain regions, including the bilateral motor cortex, bilateral putamen, bilateral thalamus, bilateral supplementary motor cortex, bilateral secondary somatosensory area, and bilateral cerebellum [41]. During recovery from Bell’s palsy, activation is observed in both the cerebellum and cerebral cortex. However, the intensity and location of the activation can vary over time and between individuals, suggesting the operation of a dynamic neural reorganization process during recovery [42].

## 3. Biomolecules Increased in the CNS after Peripheral Facial Nerve Injury

### 3.1. Prosaposin

Prosaposin (PS), a neurotrophic factor upregulated after nerve injury that is involved in nerve repair and regeneration, is known to promote the survival and growth of neurons and to enhance the production of growth factors and extracellular matrix components that are important for nerve regeneration. Accordingly, it has been studied for its potential therapeutic use in the treatment of nerve injuries and neurodegenerative diseases [43]. PS is a precursor protein that is cleaved into four smaller proteins, saposins A, B, C, and D [44]. These saposins function as coenzymes with sphingolipid activator proteins and are involved in the breakdown of sphingolipids, which are important components of cell membranes in the nervous system [45]. PS has also been shown to exert neuroprotective and neurotrophic effects on neurons and glial cells and is expressed after nerve damage as part of the process of nerve regeneration and repair [46]. PS is transported to lysosomes, where it undergoes proteolytic processing into the four saposins, which are required for normal hydrolysis of sphingolipids [47]. PS levels were found to be significantly increased in the facial nerve nucleus after facial nerve transection, suggesting the activation of various neurotrophic activities in facial nerve cells [48]. The addition of PS to collagen-filled nerve guides after sciatic nerve transection in guinea pigs was found to promote increased peripheral nerve regeneration within the guide [49]. PS administration was additionally shown to help ameliorate the atrophy of anterior spinal horn and dorsal root ganglion neurons [49]. The expression of PS mRNA was found to be increased in a rat model of focal cerebral ischemia and cortical injury, suggesting its potential role in regulating cerebral nerve regeneration [50]. PS has been shown to have protective effects on the nervous system and to play a role in the activation of G proteins. For example, the orphan G protein-coupled receptors (GPCRs), GPR37 and GPR37L1, which are expressed in neurons and glial cells in the nervous system, are thought to be involved in mediating the effects of PS [51]. Both GPR37 and GPR37L1 are known to stimulate the self-binding of prosaptide, which in turn activates signaling pathways that promote endocytosis in the nervous system [52]. In addition, small interfering RNA (siRNA)-mediated knockdown of endogenous astrocyte GPR37 and GPR37L1 was reported to attenuate the protective effects of prosaptide and PS on astrocytes [52]. GPR37 and GPR37L1 were found to be increased in microglia and astrocytes in the facial nuclei of mice following facial nerve transection [53]. Although GPR37 mainly acts in neurons, GPR37L1 is predominantly expressed in microglia or astrocytes. Increased PS in damaged neurons produces neurotrophic factors through GPR37L1, which is involved in nerve recovery [53]. Hippocampal and cortical neurons show increased immunoreactivity and expression of PS mRNA following kainic acid-induced excitotoxicity, and increased PS levels were shown to improve neuronal survival by promoting the delivery of lysosomal enzymes to damaged neurons after injury [54] (Table 1).

### 3.2. SHARPIN

Expression of shank-associated RH domain-interacting protein (SHARPIN) is upregulated in the brainstem of mice with herpes simplex virus-1 (HSV-1)-induced facial paralysis [55]. Infection of the facial nerve by HSV-1 can cause a type of facial nerve paralysis known as Bell’s palsy. Such infections usually occur in the setting of a compromised immune system, which allows the virus to reactivate and travel to nerve cells, including the facial nerve. Inflammation and immune responses to viral infection can cause damage to the facial nerve, resulting in paralysis [64]. SHARPIN, a type of linear ubiquitin chain-associated protein [65], is also a post-synaptic density protein located in the cytoplasm that serves as a key component of tumor necrosis factor (TNF)-type signaling pathways, playing important roles in inflammatory responses, cell proliferation, apoptosis, and organ development [66]. In particular, SHARPIN has been shown to promote cell survival through a TNF-dependent NF-κB anti-apoptotic signaling pathway [67]. Notably, the NF-κB signaling pathway is a crucial mediator of many physiological processes, such as cell division, cell survival, differentiation, immunity, and inflammation, and its dysregulation has been implicated in a wide range of diseases, including chronic inflammation, immune deficiency, autoimmune disorders, and cancer [68]. Consistent with a crucial role for SHARPIN in regulating activation of the NF-κB signaling pathway, SHARPIN-deficient mice also exhibit chronic inflammation and increased susceptibility to various inflammatory diseases [69,70].

### 3.3. Nitric Oxide Synthase and Nitric Oxide

Nitric oxide (NO), a free radical produced under physiological conditions, is a small signaling molecule that functions as a neuromodulator in the CNS [71]. NO has both protective and harmful effects and thus plays a complex role in the CNS. NO is produced by the enzyme nitric oxide synthase (NOS), which is expressed in both the peripheral and central nervous systems. There are three isoforms of NOS, all of which utilize L-arginine and molecular oxygen as substrates and require the cofactor, reduced nicotinamide-adenine-dinucleotide phosphate (NADPH) [72]. Inducible NOS (iNOS) is induced by inflammatory cytokines and toxins and is the only NO-synthesizing NOS form that is activated independently of calcium, producing large amounts of NO with long-lasting activity [73]. Overproduction of NO due to iNOS expression induces DNA and tissue damage and has been implicated in the pathogenesis of many diseases. While excessive production of NO can cause cytotoxicity, oxidative stress, and inflammation, leading to damage and cell death in the CNS [74], NO can also exert cytoprotective effects by regulating cerebral blood flow, neurotransmission, and inflammation. Therefore, the balance between NO production and degradation is crucial for maintaining proper neuronal function and preventing neurodegenerative diseases [75].

NO plays a critical role in both non-specific and immunological host defense and has antibacterial effects through cytotoxic or cytostatic actions against various pathogens [76]. NO produced by iNOS has beneficial effects on host defense mechanisms against bacteria [77]. However, in the case of HSV-1-associated facial nerve palsy, NO contributes to the pathogenesis of neuroviral infections and neurodegeneration [78]. During facial paralysis caused by facial nerve compression, NOS and NADPH-diaphorase activity are increased in facial motor neurons [56]. The resulting increase in NADPH-diaphorase activity contributes to the recovery of facial function by promoting axon regeneration [79]. In HSV-1-infected mice, iNOS-induced NO is overproduced in neurons, and HSV-1 increases apoptosis in the brainstem of mice [57]. Facial nerve damage caused by nerve compression leads to increased NOS activity and NO production in facial motor neurons and surrounding tissues [12]. This effect is attributable to the extensive activation of N-methyl-D-aspartate (NMDA) receptors, which mediate the effects of the excitatory neurotransmitter glutamate and have been implicated in neuronal cell death [71]. In the context of facial nerve damage, the release of NO by activation of NMDA receptors increases blood flow to the injured area, inducing local vasodilation [80].

### 3.4. Vasoactive Intestinal Peptide and Substance P

Vasoactive intestinal peptide (VIP), a peptide hormone in the gut that acts as a first responder to injury [81], is present in neurons of both the autonomic and sensory nervous systems and regulates neurotransmitter release in response to stressors. It also acts as an anti-inflammatory factor after injury and promotes neuron proliferation, survival, and axon growth [82]. In addition, VIP can help resolve acute inflammatory processes and may contribute to the prevention of chronic inflammation [83]. Changes in gut flora and biodiversity, as well as weight loss, have been observed in VIP-deficient mice, which show increased susceptibility to intestinal inflammation and inflammatory bowel disease [84]. In a spinal cord injury model, VIP was shown to inhibit the induction of TNFα and interleukin (IL)-6 in microglia, reducing neuronal cell loss around the lesion site [85]. Substance P (SP) is a tachykinin neuropeptide that acts as a neurotransmitter and neuromodulator in the CNS [86]. After peripheral facial nerve axotomy, SP and VIP expression are strongly increased in the facial nucleus, where the resulting early T cell recruitment is accompanied by increased levels of the pro-inflammatory cytokine IL-6 [58]. In the absence of IL-6 (IL-6-deficient mice), lymphocyte recruitment and axonal regeneration are reduced, and there is a decrease in CD3-positive T-lymphocyte recruitment and early microglia activation [87].

### 3.5. Fibroblast Growth Factor-2 and Glial Fibrillary Acidic Protein

Fibroblast growth factor-2 (FGF-2), a member of the fibroblast growth factor family, is known to have a variety of biological functions, including promoting cell proliferation, survival, and angiogenesis [88]. It is also known to play a critical role in tissue repair and wound healing, particularly in the skin and bone. In addition, it is involved in embryonic development and organogenesis and has been shown to play a role in cancer progression [89]. FGF-2, encoded by the FGF-2 gene, is synthesized as a precursor protein that undergoes proteolytic cleavage to yield the active form [88]. FGF-2 binds to specific cell surface receptors and activates downstream signaling pathways that regulate gene expression and cellular processes [90]. FGF-2 mRNA and protein are widely expressed in the brain, with the highest levels observed in astrocytes [91]. After injury or insult to the brain, FGF-2 expression is upregulated primarily in astrocytes, and its release from astrocytes plays an important role in promoting the survival and proliferation of neural progenitor cells [92]. A treatment strategy using Schwann cells overexpressing FGF-2, alone or in combination with passive stimulation, after facial peripheral nerve transection was shown to induce transient lateral branching by supporting axon regeneration but did not significantly improve functional recovery after facial nerve injury [93]. FGF-2 isoforms are upregulated in spinal cord neurons and sciatic nerves after peripheral nerve lesions [94]. Unilateral compression or transection of the lingual nerve was reported to increase the number of FGF-2-immunoreactive neurons and glia and increase the amount of FGF-2 present in reactive astrocytes of the lingual nerve nucleus [95].

Activated astrocytes, characterized by their expression of glial fibrillary acidic protein (GFAP), accumulate around nerve cells, where the FGF-2 they produce acts as a neuroprotective factor [96]. Axotomy increases the number of GFAP-positive astrocytes in the facial nucleus and enhances their nuclear expression of FGF-2. The resulting increase in FGF-2 in the cytoplasm of reactive astrocytes leads to enhanced secretion of FGF-2, which acts in a paracrine and autocrine manner to provide trophic support to the facial nucleus, thereby preventing Bell’s palsy [59]. Astrocyte activation in the CNS after peripheral nerve injury contributes to nerve regeneration by maintaining immune homeostasis [97], reflecting the critical role of astrocytes in restoring the blood-brain barrier, providing neuroprotection, and limiting the proliferation of inflammatory cells [98]. This supportive function of astrocytes is considered an important factor in the survival of damaged neurons in the CNS and the maintenance of synaptic plasticity and neurotransmitter release [99]. The expression of GFAP is regulated by various hormones, cytokines, and growth factors [100].

### 3.6. Sonic Hedgehog and Smoothened

In mammals, the hedgehog signaling pathway consists of three homologous genes, Desert hedgehog (Dhh), Indian hedgehog (Ihh), and Sonic hedgehog (Shh), among which Sonic hedgehog (Shh) signaling plays an important role in patterning and specifying cell function in the CNS [101,102]. Shh and its receptor, Smoothened (Smo), are upregulated in facial motor neurons of adult rats, a phenomenon that serves to restore the synaptic transmission necessary for nerve injury repair after facial neurectomy [60]. The Shh signaling pathway conveys information necessary for proper cell differentiation and is one of the key regulators of animal development [103]. During neocortex development, Shh signaling regulates intermediate progenitors to maintain neuron proliferation, survival, and differentiation in the neocortex [104]. Shh protein expression and Smo mRNA levels are upregulated in facial motor neurons after facial nerve axotomy in adult rats, and adenoviral-mediated overexpression of Shh is associated with the survival of axotomized motor neurons [60]. Shh has been shown to regulate stem cell proliferation in the adult rat hippocampus, and overexpression of Shh in the forebrain improves cognitive and motor impairment [105,106].

### 3.7. Calcitonin Gene-Related Peptide and Growth-Associated Protein-43

The molecular response of damaged motor neurons following facial nerve injury involves the upregulation of early genes followed by the expression of neuromodulatory and regeneration-related genes [107]. Calcitonin gene-related peptide (CGRP) is among the early representative genes upregulated in facial nerve nuclei, together with c-Jun and growth-associated protein-43 (GAP-43) [24]. Expression of CGRP, a 37-amino acid neuropeptide produced by splicing that is widely distributed in both peripheral and central nervous systems, is increased following facial nerve injury and is involved in axon extension [108]. An earlier study reported increased levels of CGRP in motoneurons of cat and mouse sciatic nerves 2 to 5 days after axotomy surgery [109]. However, a subsequent study found that the expression of CGRP, in addition to that of other neuropeptides (e.g., substance P, somatostatin, and cholecystokinin), was downregulated in primary sensory neurons after complete peripheral nerve axotomy [110]. In a rat facial nerve injury model, CGRP mRNA expression levels were correlated with nerve regeneration, suggesting that upregulation of CGRP in injured axons is involved in regeneration [111]. CGRP in the anterior horn of the spinal cord derived from motor neuron cell bodies has been shown to contribute to repair mechanisms involved in nerve regeneration following brachial plexus injury [112].

GAP-43, an axonal phosphoprotein expressed in granule cells of the rat hippocampus, is expressed at high levels during development and is re-induced by axonal regeneration in the CNS. GAP-43 is believed to play a key role in the regulation of axonal growth [113]. In vitro and in vivo studies have demonstrated the presence of GAP-43 immunoreactivity in Schwann cell precursors and mature, non-myelin-forming Schwann cells but not in mature myelin-forming Schwann cells [114]. After denervation, post-axonal GAP-43 expression was shown to be upregulated in almost all Schwann cells in the distal stump [115]. GAP-43 is also expressed by specific CNS glial cells in tissue culture and in vivo, indicating that its expression is not restricted to neurons [114]. Upregulation of GAP-43 in the facial nucleus following compression injury was shown to promote axon growth and regeneration of damaged nerves [61]. GAP-43 mRNA and protein are upregulated early after axonal injury and gradually decrease as the nerve recovers [116].

### 3.8. GDNF Family Receptor Alpha-1 and C-Ret

Glial cell line-derived neurotrophic factor (GDNF), a potent neurotrophic factor for damaged motor neurons, affects the survival and function of several neuronal cell populations in central and peripheral nervous systems. Its physiological actions, which include nerve regeneration effects, are mediated by a multicomponent receptor system composed of the receptor tyrosine kinase, c-ret, which binds GDNF with high affinity [117], and the ligand-binding protein GDNF family receptor alpha-1 (GDNFR-α), an orphan receptor tyrosine kinase [118]. GDNFR-α and c-ret mRNA are present in the substantia nigra and ventral tegmental regions and are found in motor neurons of the spinal cord and brainstem nuclei that innervate skeletal muscles [119]. Overexpression of GDNF after axotomy protects motor neurons from apoptosis and promotes cell survival in the CNS [120]. Although mRNA levels of the GDNFR-α ligand, GDNF, are not altered in facial motor neurons following traumatic facial nerve injury, GDNFR-α, and c-ret mRNA expression are increased after facial nerve ablation and contribute to the regeneration of damaged motor neurons by the GDNF nutritional/signaling system [62]. Thus, neurotrophic and growth factors exert survival-promoting effects on axonal motor neurons in the facial nerve axon model that largely reflect increased GDNF/GDNFR-α signaling, efficiently promoting axon regeneration in the adult state. Consistent with this, GDNF treatment after axotomy has been shown to improve the survival of injured retinal ganglion cells, midbrain dopaminergic neurons, and motor neurons in the CNS [121].

### 3.9. Brain-Derived Neurotrophic Factor and Tyrosine Receptor Kinase B

Brain-derived neurotrophic factor (BDNF) is a neurotrophin that plays a critical role in the survival and growth of neurons in central and peripheral nervous systems. BDNF can bind to two receptors: tyrosine receptor kinase B (TrkB) and the neurotrophin receptor, p75NTR [122]. Neurotrophic factors play important roles in the development, maintenance, and neuroplasticity of the nervous system [18]. After nerve transection, reactive Schwann cells in the remaining distal nerve produce a range of trophic factors, including BDNF, nerve growth factor (NGF), and neurotrophin-4 (NT-4) [123]. After facial nerve transection, BDNF mRNA is increased in brainstem nuclei and the thalamus; in addition to BDNF mRNA, BDNF protein is increased in the facial nucleus after axonal transection of the facial nerve, where it may contribute to the survival of motor neurons [63].

Mature BDNF preferentially binds to TrkB, a high-affinity receptor for BDNF, to elicit growth-promoting signals [124]. TrkB is a prototypical tyrosine kinase that dimerizes and autophosphorylates upon ligand binding [125]. Activation of TrkB by BDNF leads to the activation of several intracellular signaling pathways, including the MAPK/ERK and PI3K/Akt pathways, which promote neuronal survival, differentiation, and growth [124]. Neurotrophin signaling through TrkB is involved in protecting against axotomy-induced apoptosis and death of CNS neurons; consistent with this, the survival rate of axotomized hippocampal and motor neurons is low in TrkB^−/−^ mice [126].

## 4. Biomolecules Decreased in the CNS after Peripheral Facial Nerve Injury

### 4.1. Choline Acetyltransferase and Vesicular Acetylcholine Transporter

Cholinergic neurotransmission in the CNS relies on three proteins: choline acetyltransferase (ChAT), the high-affinity choline transporter (HAChT), and the vesicular acetylcholine transporter (VAChT) [127]. ChAT in cholinergic neurons synthesizes acetylcholine, which is then transported into synaptic vesicles by VAChT. It has been reported that VAChT is expressed in motor neurons in the rat facial nucleus and can be used as a functional marker for motor neurons [128,129]. VAChT is downregulated in damaged facial nerves, but its expression gradually increases with motor nerve recovery [25]. Axotomy of the adult rat facial nerve results in the downregulation of biomolecules associated with motoneurons, including ChAT and acetylcholine transporters [130,131]. ChAT is a functionally specific marker for neurons, such as primary motor neurons, that synthesize and release acetylcholine as a neurotransmitter. After unilateral transection or crushing of the XII nerve, the number of ChAT-expressing motor neurons is decreased compared with the non-lesioned side, but without a loss of neurons [132]. The axon subsequently extends back towards the facial muscles; when the axon reaches the target, the neuron starts expressing ChAT and is converted to an active neuron. ChAT can serve as a marker for nerve regeneration after facial nerve damage since the gradual increase in the number of active ChAT-expressing neurons leads to the recovery of facial motor function [124]. After axotomy, there is a decrease in ChAT-labeled septal cholinergic neurons but no neuronal death [133]. One week after facial nerve axotomy, ChAT immunoreactivity in the facial nucleus is significantly reduced, and after 2 months, ChAT expression increases in many motor neurons [111] (Table 2).

### 4.2. Potassium Sodium Chloride Cotransporter 2 and Gephyrin

After the transection of the facial nerve, there is a decrease in the expression of two important proteins: potassium sodium chloride cotransporter 2 (KCC2) and gephyrin. KCC2 is lost from somatic and proximal dendrite membranes of motoneurons after nerve injury, and its downregulation can cause depolarization of the chloride equilibrium potential, leading to reduced strength of post-synaptic inhibition [139]. Downregulation of KCC2 expression can also lead to axon re-expansion by increasing the intracellular chloride ion concentration and inhibiting the action of γ-aminobutyric acid (GABA) [140]. Gephyrin is a central factor in anchoring, clustering, and stabilizing GABA type A (GABAA) receptors at inhibitory synapses, and interacts with GABAergic synapses, contributing to the accumulation of GABAA receptors at post-synaptic sites [141]. In addition, it acts as an anchoring protein for glycine and GABA receptors, playing an important role in GABAergic transmission [142]. The expression of GABA in motoneurons has also been shown to decrease after facial axotomy but recovers to normal levels after 60 days [143]. In the rat facial nerve stem transection model, gephyrin was shown to rapidly detach from the axonal facial nucleus beginning on day 1, was lowest on day 8, and returned to normal by day 60 [135]. After two epineural sutures, the expression of KCC2 and gephyrin recovered at 60 days post-injury [136].

### 4.3. Glycogen Synthase

Glycogen is an important energy reserve that can be rapidly mobilized to meet increased energy demands in response to cellular stress [144]. In the nervous system, glycogen is predominantly found in astrocytes, which store and release glycogen-derived glucose to support neuronal activity and maintain energy homeostasis [145]. Glycogen synthase (GS), which converts glucose from uridine diphosphate glucose (UDP-Glc) into glycogen, is the key enzyme for glycogen synthesis [146]. In mammals, there are two isoenzymes of GS: liver-type GS, found only in the liver, and muscle-type GS, which is widely expressed in muscle and other tissues, including the brain [147]. GS levels in the facial nuclei of rats were reported to be reduced 7–14 days after facial nerve transection [130]. Motor neuronal changes and glial responses are closely linked to the energy supply system reflecting the significant energy required to support protein synthesis/DNA replication. The reduction in GS protein in severed motor neurons blocks energy-intensive glycogen synthesis, which is used for energy conservation and survival [148]. In sciatic nerve injury, glycogen phosphorylase (GP), which catalyzes the breakdown of glycogen to glucose, is increased in damaged motor neurons and can be used to generate molecules essential for survival [149]. GS plays an important role in glycogen metabolism and energy homeostasis in the nervous system, and its regulation is tightly linked to neuronal function and survival [150].

### 4.4. M2 Muscarinic Acetylcholine Receptor and Nicotinic Acetylcholine Receptor

Acetylcholine regulates neuronal differentiation during early development, and both muscarinic and nicotinic acetylcholine receptors regulate a variety of physiological responses, including apoptosis, cell proliferation, and neuronal differentiation [151,152]. Muscarinic receptors are GPCRs that mediate the response to acetylcholine released by parasympathetic nerves [153]. The m2 muscarinic acetylcholine receptor (m2MAchR) is essential for the regulation of various physiological functions, including cardiovascular function and smooth muscle contraction, through the activation of G protein-coupled endogenous potassium channels [154,155]. GPCRs bind ligands outside the cell and selectively bind and activate specific G proteins to trigger events inside the cell [156]. There are five subtypes of muscarinic acetylcholine receptors (M1R to M5R), which bind to different G proteins and play different roles in the nervous system [157]. m2MAchR mRNA is located in facial motor neurons and is considered a motor neuron marker.

Nicotinic acetylcholine receptors (nAChRs) have a variety of functional roles in the central and peripheral nervous system and are critical for sympathetic transmission [158]. nAChRs in motor neurons function not only as post-synaptic receptors on dendrites or cell bodies but also as pre-synaptic receptors or autoreceptors on axon terminals of neuromuscular junctions. After axotomy, mRNA for the α3 subunit was reported to be reduced in facial motoneurons of the rat, an effect that was associated with a decrease in ChAT [138]. Postganglionic dissection by cervical ganglion axotomy has been shown to reduce mRNA transcripts for α3, α5, α7, and β4 nAChR subunits and protein expression of α7 and β4 subunits [159]. Most signals are transduced by nAChRs containing the α3 subunit, a major component of nAChRs that is abundantly expressed at the mRNA level in brainstem motor nuclei, including trigeminal nuclei and facial nuclei [160]. nAChRs containing the α7 subunit contribute to the regulation of microglial activity by inhibiting the synthesis of pro-inflammatory molecules and protecting neurons [161].

### 4.5. Oligodendrocyte Myelin Glycoprotein

Oligodendrocyte myelin glycoprotein (OMgp), which is expressed in neurons and oligodendrocytes in the CNS, is a membrane-anchored protein tethered by a glycosylphosphatidylinositol moiety [162,163]. Myelin-related glycoproteins and Nogo expressed on oligodendrocytes bind to the Nogo receptor to inhibit neurite outgrowth, and their expression in oligodendrocytes inhibits axon regeneration after CNS injury. OMgp expression was reported to be downregulated in the facial nucleus at 5–7 days (mRNA) or 5–14 days (protein) after facial nerve transection, returning to control levels at 28 days after axotomy [134]. This indicates that the decrease in OMgp expression is attributable to its downregulation in facial motor neurons rather than oligodendrocytes and that the change in neuronal OMgp expression is likely involved in the reconnection of neural circuits between axonal facial neurons and upper motor neurons after amputation [134].

### 4.6. GABAA and GABAB Receptors

Facial neurons receive strong GABAergic innervation and are endowed with numerous GABAA and GABAB receptors. GABAA receptors (GABAARs) are ligand-gated chloride channel complexes formed from receptor subunits classified into four families-α (1–6), β (1–3), γ (1–5), and δ-according to sequence similarity [164]. In the mammalian brain, they primarily provide rapid inhibition, mainly as αβγ, αβδ or ρ heteromeric combinations of pentamers [165]. Excitatory neurotransmission is increased after facial nerve axotomy in association with a decrease in GABAA expression in the cell body of motor neurons in facial nuclei, reflecting downregulation of α1, β2, and γ2 mRNAs [135]. This results in changes in the properties, or a reduction in the synaptic transmission of GABAergic inputs. Changes in Schwann cell-axon connections provide a signaling mechanism to the cell body to downregulate the α1 subunit [166]. Transection of rat facial nerves was reported to cause downregulation of GABAAR α1 protein levels in injured motor neurons from 5 days to 5 weeks post-injury [167]. This decrease in GABAAR α1 protein levels in the damaged nucleus did not result from motor neuron apoptosis but instead was attributable to changes in protein levels of glutamate decarboxylase, vesicular GABA transporter, and GABA transporter-1 in GABAergic neurons.

GABAB receptors belong to the superfamily of seven transmembrane domain-containing GPCRs and are linked to Ca^2+^ and K^+^ channels by G protein and second messenger transduction pathways. After facial nerve axotomy, GABAB receptor levels were reported to increase in the facial nucleus [168]. However, it has also been reported that the abundance of mRNA for GABA (B1B) and GABA (B2) subunits is reduced in motoneurons after facial nerve axotomy, in association with a corresponding change in protein levels of the GABA (B2) subunit, but not the GABA (B1B) subunit [135].

### 4.7. α-Amino-3-Hydroxy-5-Methylisoxazole-4-Propionic Acid Receptor and N-Methyl-D-as Partate Receptor

Facial motor neurons receive inputs from various sources, including premotor neurons from the trigeminal nucleus and glutamate nerve endings from the sublingual nucleus and reticular body [169]. These inputs are mediated by glutamate receptors, including α-amino-3-hydroxy-5-methylisoxazole-4-propionic acid (AMPA) receptors and *N*-methyl-D-aspartate (NMDA) receptors [170]. AMPA and NMDA receptors are ionotropic glutamate receptors that play important roles in synaptic transmission and plasticity in the nervous system [171]. In the case of facial motor neurons, the glutamate receptor subunits, GluR2 and GluR3, of AMPA receptors were found to be reduced after facial nerve axotomy. This reduction in GluR2 and GluR3 subunits in motor neurons can lead to an increase in intracellular Ca^2+^ concentration, which may be excitotoxic to neurons [136]. On the other hand, NMDA receptors, composed of NR1 and NR2 subunits, are involved in synaptic development and plasticity [172]. The downregulation of the NR1 subunit in facial motor neurons after nerve injury and the detection of NR2A and NR2B subunits in their cell body may have an impact on the plasticity and function of these neurons, which are crucial for motor control and recovery following injury [173].

### 4.8. Vesicular Glutamate Transporter

Glutamate, the main excitatory neurotransmitter in the brain, is transported into synaptic vesicles by three types of vesicular glutamate transporters (VGLUTs): VGLUT1, VGLUT2, and VGLUT3 [174]. These transporters are responsible for packaging glutamate into vesicles in the pre-synaptic terminal; from here, it can then be released into the synaptic cleft and bind to glutamate receptors on the post-synaptic membrane, leading to excitatory synaptic transmission [175]. VGLUT2 is expressed in canonical glutamatergic neurons. VGLUT1 and VGLUT2 expression was reported to be reduced in the lumbar dorsal root ganglia of rats following sciatic nerve axotomy [176], which was also reported to reduce VGLUT1 immunoreactivity in the spinal cord [177]. After facial nerve transection, VGLUT2 is reduced in the facial nucleus, as evidenced by a large decrease in VGLUT2 staining [136]; this reduction, which likely serves to protect facial motor neurons from excitotoxic effects, is associated with the upregulation of glutamate transporters in activated microglia in facial nuclei [178].

### 4.9. Post-Synaptic Density-95 and Carboxy-Terminal PDZ

Post-synaptic density-95 (PSD-95), a scaffolding protein localized to the post-synaptic density, plays a key role in signal transduction, synaptic plasticity, and synaptogenesis and facilitates NO synthesis by clustering NMDARs on synaptic membranes and binding to neuronal NOS (nNOS) [179]. Carboxy-terminal PDZ (CAPON), an nNOS-binding protein, competes with PSD-95 for interactions with nNOS, thus preventing the association of NMDARs with nNOS. PSD-95 and CAPON mRNAs in facial motor neurons decrease from their initial expression level from 7 to 14 days after facial nerve axotomy and then gradually increase and recover to control levels by postoperative day 35 [137]. The recovery of synaptic function, measured as an increase in nNOS expression, precedes the recovery of PSD-95 and CAPON mRNA expression.

## 5. Central Facial Nerve Biomolecules and Processes Involved in Peripheral Facial Nerve Damage

### 5.1. Autophagy

Autophagy, a cytoprotective process commonly found in eukaryotes, plays a critical role in maintaining cellular and tissue homeostasis by removing damaged organelles, pathological proteins, and dysfunctional macromolecules from within cells [180]. Autophagy regulates various physiological and pathological processes through the lysosomal degradation pathway, including nerve regeneration, myelin development, myelin degradation, and neurodegeneration [181]. Autophagy is an initial process activated after facial nerve injury that serves to repair the damage [182]. Autophagy in Schwann cells has been found to be beneficial for scar reduction and myelination, playing a crucial role in preventing or delaying the onset and chronicity of neuropathic pain and neuropathy [183]. Activators of autophagy were shown to be effective in promoting nerve regeneration and motor recovery in a sciatic nerve crush model [184]. For example, CXCL12 increased the autophagy markers, LC3II/I, indicative of autophagy activation, in association with a reduction in mTOR phosphorylation after facial nerve injury, improving facial nerve function and myelin regeneration [185]. Treatment of facial nerve injury with basic fibroblast growth factor (bFGF) restored the morphology and function of the damaged facial nerve by promoting autophagy and inhibiting apoptosis through activation of the P21-activated kinase 1 (PAK1) signaling pathway [186] (Table 3).

### 5.2. Reactive Oxygen Species

Reactive oxygen species (ROS) are known to contribute to the pathogenesis of facial nerve injury [189]. These highly reactive compounds can damage cell membranes and organelles and are involved in ischemia-reperfusion injury [190]. ROS can take the form of free radical species such as NO, which contain unpaired electrons, and non-radical species, including superoxide and hydrogen peroxide. Both radical and non-radical species are produced during normal metabolic processes and can be beneficial in small amounts. However, excessive production can lead to tissue damage, accompanied by the production of oxidants [191]. Tissue damage caused by ischemia or trauma can accelerate free radical reactions and increase oxidative stress. Antioxidants such as superoxide dismutase (SOD) that scavenge free radicals can be used to inhibit oxidative stress in nerve cells and reduce nerve damage. In this context, SOD treatment after ischemic peripheral facial paralysis was shown to reduce facial nerve damage and affect facial motor regeneration [24]. Herpesvirus 7 (HHV7) infection, which causes Bell’s palsy, increases oxidative stress by producing ROS in facial nerve injury. Expression of the cytochrome C oxidase (COX) isoform, complex IV subunit 4 isoform 2 (Cox4i2), is increased in Schwann cells infected with HHV7, resulting in increased COX activity, ROS production, and apoptosis. Such oxidative damage-induced apoptosis can be ameliorated by treatment with malondialdehyde (MDA), SOD, or glutathione (GSH) [187].

### 5.3. Interleukin-10

Interleukin-10 (IL-10) is a cytokine that plays a crucial role in regulating inflammation and immune responses [192]. In the CNS, IL-10 is upregulated in a number of pathological contexts, such as cerebral artery occlusion, excitotoxicity, and traumatic brain injury [193]. Astrocytes and microglia are potential sources of IL-10 production, and IL-10 receptors are expressed in microglia, astrocytes, oligodendrocytes, and neurons [194]. The inflammation- and immune-regulatory functions of IL-10 are critical for the survival of facial motor neurons following facial nerve injury [195]. Consistent with this, mice with selective knockout of IL-10 exhibit decreased facial nerve survival compared to wild-type mice [196]. IL-10 produced by neurons and astrocytes plays an important role in maintaining neuronal cell homeostasis and providing neuroprotective nutrition after axotomy [197]. After facial nerve axotomy, IL-10 plays a vital role in maintaining an anti-inflammatory environment in the CNS and is produced by several other cells, including T helper 2 (Th2) cells, to exert a direct anti-apoptotic effect on neurons [196]. After facial nerve injury, transgenic mice overexpressing IL-10 (GFAP-IL-10Tg mice) displayed a higher density of facial nerve microglia with higher levels of IL-10 mRNA expression compared with wild-type mice. The resulting overproduction of IL-10 protects against facial nerve injury by acting as a neurotrophic factor and preventing neuronal cell death [26].

### 5.4. Calcium

Calcium is involved in several signaling pathways that regulate cellular homeostasis [198]. Calcium signaling is critical to the glial environment of neurons and plays an important role in neuron survival and regeneration after axonal injury [199]. The expansion of excitotoxic glutamate resulting from stroke, epilepsy, or traumatic brain injury leads to an elevation in intracellular calcium ions that, if excessive, can trigger processes leading to neuronal cell death and necrosis [200]. Elevated cytosolic calcium levels have been found in axons and soma of mechanoreceptor neurons within minutes after axotomy, indicating the importance of calcium signaling in response to axonal injury [201]. Disrupting calcium influx may be useful in counteracting injury-induced neuronal cell death [200]. Nimodipine has been shown to have neuroprotective effects after various lesions in the CNS, including in the facial motor nucleus after intracranial transection of the facial nerve [188]. Thus, nimodipine and other calcium antagonists may be useful in protecting neurons from injury-induced cell death and promoting regeneration after axonal injury [202].

## 6. Changes in the CNS Resulting from Facial Nerve Injury

Cortical plasticity, which refers to the cortex’s ability to adapt to a changing environment and new information, occurs in brain and peripheral nerve lesions [34]. Cortical reorganization plays an important role in the recovery of patients with Bell’s palsy, with fMRI showing that it is associated with increased functional connectivity in the anterior cingulate cortex in patients recovering from facial palsy [38]. An analysis of changes in facial nerve morphology 4 weeks after facial nerve transection revealed changes in electrophysiological properties and firing frequency adaptation in two types of putative facial nucleus motoneurons [203]. Specifically, it was shown that firing rates are reduced, and firing patterns are altered in motoneurons in the context of facial nerve palsy and neurotmesis due to facial nerve neuropathy. This suggests that peripheral facial nerve injury can cause changes in the electrophysiological properties and firing frequency adaptation of motoneurons in the facial nucleus [10]. MRI of patients with congenital facial palsy, clinically defined as facial palsy of the seventh cranial nerve present at birth or immediately after birth, revealed abnormalities in the facial nucleus, Willis circle, and corpus callosum [29]. After recovery from facial nerve palsy, neurons innervating the zygomatic muscle, identified using retrograde staining with horseradish peroxidase (HRP), were located in the facial nucleus; labeled neurons were also found in the facial nucleus region following injection of HRP into the orbicularis, zygomatic, and orbicularis oris muscles [204,205]. In a monkey model, blocking the facial nerve by inserting a needle into the facial nerve trunk resulted in changes in facial nucleus nerve cells [206]. Collectively, these observations indicate that peripheral facial nerve injury can lead to changes in the CNS, including cortical and functional plasticity, alterations in the firing patterns of motoneurons, and abnormalities in the facial nucleus, Willis circle, and corpus callosum (Table 4).

## 7. Discussion

The changes following facial nerve injury highlighted here can trigger cascades of events in the CNS, including alterations in neurotransmitter release, glial activation, and changes in gene expression, all of which can contribute to functional and structural plasticity (Figure 1). The brain can adapt to these changes through reorganization and rewiring of neural circuits, which can help to restore function and improve recovery from facial paralysis. Understanding the mechanisms underlying these changes in the CNS can provide insights that aid in the development of novel therapeutic interventions targeting the central pathways involved in facial nerve regeneration and recovery.

One critical process in maintaining cellular and tissue homeostasis is autophagy, which removes damaged organelles, pathological proteins, and dysfunctional macromolecules from within cells. Macrophages are crucial players in nerve injury and regeneration. After nerve injury, macrophages trigger an inflammatory response and help to clear and regenerate damaged tissue [207]. Moreover, the coordinated actions of the M1 and M2 subtypes can establish a favorable microenvironment for the release of cytokines that aid in the repair of damaged tissue [208]. As a result, macrophages play a critical role in both nerve damage and regeneration. After facial nerve injury, activation of autophagy is beneficial for reducing scarring and promoting myelination, which facilitates regeneration and motor recovery [183]. ROS levels also increase significantly after peripheral nerve injury, leading to cellular damage through oxidative stress [209]. Thus, inhibiting ROS production could mitigate oxidative damage induced by facial nerve injury. Moreover, IL-10 plays a significant role in regulating inflammatory and immune responses following facial nerve injury [196]. Inhibition of IL-10 reduces facial nerve survival, whereas overproduction of IL-10 protects against facial nerve injury by preventing neuronal cell death.

After peripheral facial nerve injury, activated PS and SHARPIN act on neurons and glial cells in the CNS to support cell survival [48,55]. Additionally, the increased release of NO promotes circulation at the damaged site by inducing local vasodilation [80]. iNOS-mediated NO production also has a beneficial effect on the host’s defense mechanisms against bacteria, but excessive increases in NO can lead to cytotoxicity and cell death. As first responders to injury, VIP and PS help resolve acute inflammatory processes and inhibit the induction of TNFα and IL-6 in CNS microglia, which reduces neuronal loss around the lesion site [58]. FGF-2 plays an important role in tissue development and damage repair, increasing the number of GFAP-positive astrocytes in the facial nucleus after facial nerve injury and enhancing their expression of FGF-2 [59,93]. FGF-2 promotes the survival of injured neurons and preserves synaptic plasticity and neurotransmitter release, which are crucial for neuroprotection [88]. Shh and Smo, which plays a regulatory role in transmitting information necessary for cell differentiation, are upregulated in facial motor neurons, where their increased activity serves to restore the synaptic transmission necessary for nerve repair after facial neurectomy [60]. CGRP and GAP-43, derived from motor neuron cell bodies, are involved in nerve regeneration repair mechanisms; their activity promotes the regeneration of damaged nerves by positively affecting axon regeneration and growth [112,114]. Overexpression of GDNF protects motor neurons from apoptosis and promotes cell survival [120]. Activation of its GDNFR α/c-ret receptor complex contributes to facial nerve regeneration in motor neurons in the spinal cord and brainstem nuclei, thereby promoting axonal regeneration [119]. BDNF and TrkB are increased in the facial nucleus after facial nerve injury and help maintain CNS homeostasis by preventing neuronal cell death [63]. These factors and signaling pathways all work together to promote tissue repair and recovery after facial nerve injury.

After facial nerve axotomy, cholinergic neurotransmitters in the facial nucleus of the CNS are downregulated, and the number of ChAT neurons, the primary motor neurons involved in acetylcholine synthesis, is reduced [130]. However, the expression of ChAT increases with the recovery of facial motor function and can be used as a functional marker of nerve regeneration. Furthermore, a decrease in m2MAchR indicates a decrease in motor neurons, and a decrease in nAChR stimulates the production of pro-inflammatory molecules [138].

In addition, the excitatory neurotransmitter glutamate increases after facial nerve transection, as does the expression of GABAA, KCC2, and gephyrin, indicating a decrease in inhibitory neurotransmission [135,143]. However, GABAB receptor levels are increased, notably in microglial cells of the facial nucleus [168]. Following facial nerve transection, GS levels are reduced in the facial nucleus, thereby disrupting the synthesis of glycogen, an essential molecule used for energy conservation and survival [130]. OMgp is expressed in neurons and oligodendrocytes, and its expression in oligodendrocytes inhibits axon regeneration after central nervous system injury. After nerve injury, OMgp expression was found to decrease on day 14 and return to normal levels after day 28 [134]. This expression pattern was not observed in oligodendrocytes but was observed in neurons. This suggests that the decrease in OMgp after facial nerve injury is not related to changes in the expression of oligodendrocytes but rather to changes in the expression of neurons and implies that changes in OMgp expression in neurons are involved in the reconnection of the neural circuit between axonal facial neurons and upper motor neurons after injury [134]. The reduction in AMPA and NMDA receptors in the axotomized facial nucleus may result in the excitotoxicity of facial motor neurons owing to increases in intracellular calcium concentration [136]. An excessive increase in calcium causes neuronal cell death and necrosis, whereas reducing calcium influx after facial nerve injury may increase neuronal survival.

The present review of the biomolecules and processes that change in the CNS after facial nerve injury provides insight into how peripheral facial nerve injury affects the CNS and suggests strategies for harnessing this information to promote recovery from injury. In order to treat patients with facial nerve damage, it is necessary to control the signal transmission system of the central facial nerve and/or to discover and develop a control agent or a nerve regeneration agent. Studying changes in the biomolecules involved in facial nerve injury should enable the identification of the factors that play important roles in the functional recovery of facial nerve damage, which can then be exploited in clinical treatment. The results summarized in this review have implications for various treatments, including facial nerve damage surgery and drug treatment for facial nerve injury.

## Figures and Tables

**Figure 1 antioxidants-12-01036-f001:**
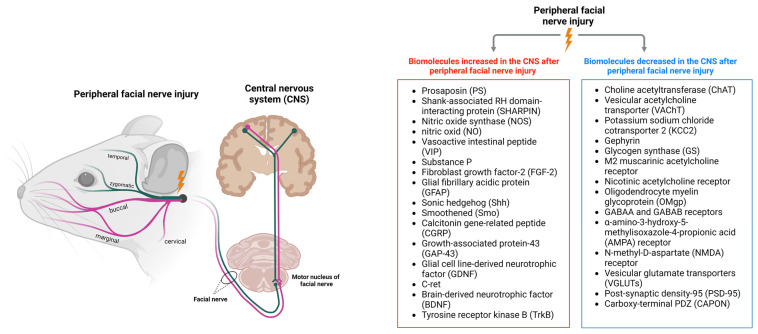
Biomolecules increased and decreased in the CNS after peripheral facial nerve injury.

**Table 1 antioxidants-12-01036-t001:** Biomolecules increased in the CNS after peripheral facial nerve injury.

Biomolecules	Reference	Animal Model	Surgical Procedures	Experimental Design	Evaluations	Results	Conclusions
Prosaposin	Kunihiro et al., 2020 [53]	Male Wistar rats (*n* = 16)	Left facial nerve transection	-Group 1: Postoperative day 1 *(n =* 4) -Group 2: Postoperative day 3 *(n =* 4) -Group 3: Postoperative day 7 *(n =* 8)	-Immunohistochemistry -Immunofluorescence	-GPR37-IR was more intense in the cytoplasm of motoneurons on the operated side than on the untreated side and was markedly higher in microglia and astrocytes on the operated side. -Microglia with a strong GPR37L1-IR signal covered the damaged neurons.	-At the same time, secreted PS stimulates microglia or astrocytes via GPR37L1 to produce neuroprotective factors that protect the damaged neurons.
SHARPIN	Li et al., 2015 [55]	Balb/c mice *(n =* 142)	Inoculation on the right side with 25 µL of HSV-1 solution (2 × 10^7^ TCID_50_/mL)	-Group 1: HSV-1 *(n =* 108) Group 1A: HSV-1 (sacrificed at 6 h and 1, 2, 3 and 7 days) Group 1B: HSV-1 +MPSS (30 mg/kg; for 2 days) Group 1C: HSV-1 + MPSS + RU486 (20 mg/kg) -Group 2: Normal saline *(n =* 25) -Group 3: Normal control *(n =* 9)	-Real-time PCR -Western blot -Immunofluorescence	-SHARPIN mRNA and protein expression were distinctly increased and peaked at 2 days after facial paralysis, then declined to normal levels during the following 5 days. -On the second day post-HSV-1 inoculation, the detection of SHARPIN protein by immunofluorescence coincided with peak SHARPIN expression. -SHARPIN expression was detected in the brainstem of HSV-1-infected mice and was localized to the cytoplasm of the facial nerve nucleus in the brainstem.	-The enhanced activity of SHARPIN in the early phase represents an important mechanism in HSV-1-induced facial paralysis. -SHARPIN might be a new target for the treatment of HSV-1-induced facial paralysis.
NOS	Wong et al., 1995 [56]	Male Wistar rats *(n =* NA)	Left facial nerve compression	-Group 1: Left facial nerve compression -Group 2: Sham-operated	-NOS assay -Histochemistry	-NOS activity in the FMN and surrounding tissues increased markedly (by ~60%) on the same side within 5 days after compression of the facial nerve, and remained significantly increased up to day 20. NOS activity decreased to control levels by day 30 after compression. -A significant decrease in NOS activity was observed on day 40. -NADPH-diaphorase reactivity in the FMN was markedly increased between days 21 and 42, peaking on day 35.	-Endothelial NOS activity increased in the initial period after nerve compression, coinciding with the period of facial paralysis and possible neuronal damage and perikaryal reactions in the FMN.
iNOS	Mao et al., 2012 [57]	Balb/c male mice *(n =* 143)	Inoculation on the left side with 25 µL of HSV-1 solution (2 × 10^7^ TCID_50_/mL)	-Group 1: HSV-1 *(n =* 110) Group 1A: HSV-1 at different time points (6 h and 1, 2, 3, and 7 days) Group 1B: HSV-1 +MPSS (30 mg/kg; for 2 days) Group 1C: HSV-1 + MPSS + RU486 (20 mg/kg) -Group 2: Sham-normal saline *(n =* 24) -Group 3: Normal control *(n =* 9)	-Hematoxylin and eosin (H&E) staining -Trichrome staining -RT-PCR -Western blot -Immunohistochemistry	-HSV-1 increased mRNA expression of iNOS in the brainstem of facially paralyzed mice. -The expression of iNOS was increased and peaked at 2 days post-induction of facial paralysis, and then declined to normal levels during the following 5 days. -iNOS expression in motor neurons and glial cells was upregulated dramatically after HSV-1 infection.	-The nerve cell damage induced by HSV-1 infection was related to the overproduction of NO by iNOS. -Enhanced activity of the gene encoding iNOS in the early phase represents an important mechanism in HSV-1–induced facial paralysis.
VIP SP	Mignini, Fiorenzo et al., 2012 [58]	Male Wistar rats *(n =* 30)	Right facial nerve resection	-Group 1: Nerve axotomy *(n =* 30); 7, 14, or 21 days -Group 2: Control left side *(n =* 30)	-Immunohistochemistry -RT-PCR	-VIP^+^ and SP^+^ cells increased for 7 to 14 days after axotomy, whereas CD3^+^ cells increased from 48 h after axotomy and peaked at day 14 after injury. -VIP and SP mRNAs were as upregulated in the axotomized side 7 and 14 days, respectively, after surgery. -IL-6 levels 48 h after axotomy were significantly higher than those at 24 h.	-Actions of the neuropeptides, VIP and SP, are beneficial in inflammation -VIP and SP expression in the facial nerve could explain the role of T cells in preventing initial neuronal death or slowing the rate of neurodegeneration and neuronal loss.
FGF-2 GFAP	Coracini, Karen F. et al., 2010 [59]	Male Wistar rats *(n =* 18)	Right facial nerve crush for 30 s or a 3 mm transection	-Group 1: Facial nerve crush *(n =* 6) -Group 2: Facial nerve transection *(n =* 6) -Group 3: Sham-operated *(n =* 6)	-Immunohistochemistry	-The vast majority of nuclear FGF-2 immunoreactivity was associated with GFAP-positive astrocytes in rat facial nuclei. -A higher amount of FGF-2 was found in nuclei of reactive astrocytes of axotomized facial nuclei. -The degree of astroglial activation and the magnitude of changes in astroglial FGF-2 immunoreactivity were greater after facial nerve transection (without fiber regeneration) than after crush injury.	-The presence of FGF-2 immunoreactivity in neurons and astrocytes of the facial nucleus indicates that FGF-2 may be an important growth factor for peripheral motoneurons. -Expression of astroglial/neuronal FGF-2 in the facial nucleus may be correlated with local paracrine/autocrine trophic effects on axotomized facial motoneurons.
Shh Smo	Akazawa et al., 2004 [60]	Wistar rats *(n =* NA)	Facial nerve transection	-Group 1: Control -Group 2: Axotomy -Group 3: Axotomy + AdV-Shh -Group 4: Axotomy + AdV-lacZ	-Immunofluorescence -Immunohistochemistry -Northern and Western blot	-Shh expression was upregulated beginning 24 h after axotomy and declined at 4 weeks. -Smo mRNA expression was upregulated at 24 h after axotomy. -Shh transcripts and polypeptides were not upregulated after axotomy of neonatal rats.	-Shh is identified as a key molecule in nerve regeneration and shown to play a regulatory role after nerve injury. -Shh has potential therapeutic applications for regeneration of neuronal tissues after injuries in vivo.
CGRP GAP-43	Mohri et al., 2001 [24]	Male Sprague-Dawley rats *(n =* 48)	-Transient paralysis (ischemia) -Right facial nerve transection	-Group 1: Right side facial nerve injury *(n =* 36) -Group 2: Untreated left side, serving as a Control *(n =* 36) -Group 3: Axotomy + saline *(n =* 3) -Group 4: Axotomy + SOD *(n =* 3) -Group 5: Ischemia + saline *(n =* 3) -Group 6: Ischemia + SOD *(n =* 3)	-Confocal laser-scanning microscopy -Immunohistochemistry	-CGRP mRNA levels in Group 1 showed a first peak at postoperative day 3 and a second peak on postoperative day 14. -The first increases in CGRP mRNA expression in Group 5 were less than those in Group 3. -The time course of c-Jun mRNA expression following ischemic nerve injury was similar to that after axotomy, although axotomy produced a greater upregulation of c-Jun mRNA than ischemia. -GAP-43 mRNA levels returned to control values before postoperative day 14. -CGRP mRNA expression in Group 6 on postoperative day 3 was inhibited compared with that in Group 5	-CGRP and c-Jun mRNA expression may be dependent upon the extent and severity of nerve damage. -A minor injury to the peripheral nerve may elicit a small regenerative change in the cell body. -Free radicals generated by ischemia may be partially responsible for ischemic nerve damage and changes in gene expression in motoneurons.
Shh GAP-43	Ni et al., 2020 [61]	Male Wistar rats *(n =* 50)	Right facial nerve transection	-Group 1: Axotomy *(n =* 10) -Group 2: Reinjury involving chronic axotomy *(n =* 40); at 12, 20, 28, and 36 weeks after the initial facial nerve axotomy	-Immunohistochemistry -Toluidine blue staining -Transmission electron microscopy -RT-PCR -Western blot	-Following reinjury, GAP-43 mRNA and protein in facial motoneurons were initially upregulated, but then gradually decreased. -Strong Shh immunoreactivity was observed in the cell bodies of facial motoneurons (GAP43-positive cells), but was not detected in the cell bodies of astrocytes (GFAP-positive cells). -Shh protein expression decreased over time following facial nerve reinjury.	-The regeneration potential of the facial nerve peaks within 5 months after chronic facial nerve axotomy in rats and may be dependent on activation of the Shh signaling pathway.
GDNFR-α c-ret	Burazin et al., 1998 [62]	Male Sprague-Dawley rats *(n =* 24)	Right facial nerve resection or crush	-Group 1: Nerve axotomy; at 1, 3, 7, 14 or 21 days *(n =* 3–4) following surgery -Group 2: Sham-operated; at 1 or 21 days *(n =* 2) following surgery	-In situ hybridization	-c-ret mRNA increased 1.4-fold in the ipsilateral facial nucleus 1 and 3 days following unilateral facial nerve crush or resection, respectively, but returned to levels equivalent to those on the contralateral side by postoperative days 7–21. -GDNFR-α mRNA was increased 2- to 3-fold in the ipsilateral facial nucleus at 1 and 3 days after facial nerve crush, reaching levels similar to those 3–21 days after resection	-The GDNF signaling system exerts powerful and long-lasting trophic effects in damaged neurons, further suggesting the broad potential for biological and therapeutic actions of GDNF and related factors in the CNS, particularly on motor neurons.
BDNF TrkB	Kobayash et al., 1996 [63]	Male Sprague-Dawley rats *(n =* 90)	Left facial nerve transection	-Group 1: Nerve axotomy *(n =* 90); at 3, 8, 16 and 24 h, and 2, 3, 4, 7, 14 and 21 days *(n =* 10) -Group 2: Control-right facial nerve *(n =* 90)	-In situ hybridization -RT-PCR -Western blot	-Axotomy increased BDNF mRNA expression in axotomized facial motoneurons as early as 8 h after injury and sustained it at levels 2- to 4-fold higher than those on the contralateral side for several days. -Increased expression of BDNF mRNA and protein was followed by increased expression of TrkB mRNA encoding the BDNF receptor, starting 2 days after axotomy and persisting for 2–3 weeks. -Axotomy increased both BDNF mRNA and protein several folds in facial motoneurons, as demonstrated by their cellular localization.	-Upregulation of BDNF mRNA within axotomized facial motoneurons and the production of BDNF protein within facial motor nuclei argue for autocrine trophic support of injured motoneurons. -BDNF increases might contribute to the survival of motoneurons after target disconnection by axotomy.

Abbreviations: GPR37-IR: G protein-coupled receptor 37; SHARPIN: shank-associated RH domain-interacting protein; HSV-1: herpes simplex virus-1; NOS: nitric oxide synthase; FMN: facial motor nucleus; NADPH-diaphorase: nicotinamide-adenine-dinucleotide phosphate-diaphorase; iNOS: inducible NOS; NO: nitric oxide; VIP: vasoactive intestinal peptide; SP: substance P; CD3: cluster of differentiation 3; IL-6: interleukin-6; FGF-2: fibroblast growth factor-2; GFAP: glial fibrillary acidic protein; Shh: sonic hedgehog; Smo: smoothened; CGRP: calcitonin gene-related peptide; GAP-43: growth-associated protein-43; GDNF: glial cell line-derived neurotrophic factor; CNS: central nervous system; BDNF: brain-derived neurotrophic factor; TrkB: tropomyosin receptor kinase B.

**Table 2 antioxidants-12-01036-t002:** Biomolecules decreased in the CNS after peripheral facial nerve injury.

Biomolecules	Reference	Animal Model	Surgical Procedures	Experimental Design	Evaluations	Results	Conclusions
ChAT Gephyrin KCC2	Kim et al., 2018 [111]	Male C57BL/6J mice (*n* = 42)	Transection of the main trunk of the right facial nerve except for the supraorbital nerve	Observed at days 3, 7, 14, 21, 28, and 60 after operation (7 mice per time point)	-Immunohistochemistry	-Only galanin expression had returned to normal levels 1 month after surgery. In contrast, expression of the other four molecules returned to normal levels by postoperative day 60. -Galanin appeared within cell bodies after surgery and had disappeared by day 28. -The number of ChAT-positive neurons in facial nuclei was lowest at day 7 and gradually increased after day 14, and the time course of changes in the ratio of the number of ChAT-positive neurons paralleled facial motor function. -Expression of gephyrin and KCC2 decreased from day 3 to day 28 and both recovered to normal levels by day 60; the time course of their restorations paralleled the recovery of FMN function. -Micro-separations comprising irregular spaces or astroglial processes were observed between motor neurons and pre-synapses.	-ChAT and KCC2 expression change during regeneration and may be objective indicators of regenerating axons, whereas galanin may be a marker for injured axons. -Decreases in KCC2 may play a role in re-extension of injured axons, and decreases in ChAT-positive neurons may be related to functional recovery.
ChAT VAChT Glycogen synthase	Takezawa et al., 2014 [130]	Male Wistar rats *(n =* 130)	Right facial nerve transection	Divided at various time points (1, 3, 5, 7, 14, 21, 28, and 35 days)	-Immunoblotting -Immunohistochemistry -Immunofluorescence -Nissl staining	-ChAT and VAChT in the operated nucleus were downregulated between days 3 and 14 post-injury. -GS levels in the injured nucleus were reduced beginning on day 7 post-injury, ultimately decreasing to ~50% of initial levels by postoperative day 14. -GS protein expression in motoneurons was significantly decreased in the injured nucleus. -The decrease in GS levels in injured motoneurons observed at 2 weeks post-insult recovered to original levels by 5 weeks in association with restoration of motoneuron functions.	-GS levels are altered in injured motoneurons in a rat facial nerve axotomy model. -The molecular mechanisms underlying energy metabolism in motoneurons are regulated during injury and regeneration.
ChAT VAChT m2MAchR	Ichimiya et al., 2013 [25]	Male rat littermates *(n =* NA)	Right facial nerve transection	Decapitated at early time points (1, 3, 5, 7, and 14 days) or later time points (3, 4, and 5 weeks)	-Immunoblotting -Immunohistochemistry -Immunofluorescence	-ChAT protein levels in the ipsilateral nucleus dropped significantly at an early stage after facial nerve transection in association with a parallel drop in VAchT protein levels. -ChAT and VAChT levels in the axotomized facial nucleus were downregulated starting 1 day after transection and remained depressed for 14 days. -m2MAchR levels in the transected nucleus were largely sustained (82%) up to 3 days after insult, but dropped markedly at 5 days and remained low thereafter. -ChAT and VAChT levels in the transected facial nucleus returned to control levels 4–5 weeks after injury. However, m2MAchR levels in the ipsilateral nucleus had not returned to control levels even at 5 weeks after insult.	-ChAT and VAchT were found to be downregulated in transected motoneurons almost immediately (days 1–3) after injury, whereas m2MAchR levels decreased starting on day 5 after insult. -Although m2MAchR was sustained at low levels for 5 weeks after injury, ChAT and VAChT recovered in the later stage (weeks 4–5).
OMgp	Koyama et al., 2008 [134]	Male Sprague-Dawley rats *(n =* 18)	Left facial nerve axotomy	-Group 1: Left facial nerve axotomy *(n =* 18); 1, 3, 5, 7, 14 and 28 days after nerve transection -Group 2: Sham operated-right facial nerve *(n =* 18)	-RT-PCR -Western blot -Immunohistochemistry	-OMgp mRNA levels significantly decreased from 3 to 14 days following axotomy, then returned to control levels at 28 days after transection. -OMgp immunoreactivity on the surface membrane of both neuronal soma and dendrites decreased significantly after axotomy 3–14 days after transection and then returned to control levels at 28 days post-axotomy. -OMgp expression did not change in oligodendrocytes and was not detectable in astrocytes.	-Neural OMgp might be involved in reconnecting neural circuits between axotomized and upper neurons. -Downregulation of neuronal OMgp expression after peripheral nerve axotomy provides important insights into the mechanism of OMgp in reconnecting the disconnected neural circuit after axotomy.
GABA_A_ and GABA_B_ receptors	Vassias et al., 2005 [135]	Male pigmented Long-Evans rats *(n =* 72)	-Left facial nerve cut	-Group 1: Left nerve axotomy *(n =* 42); divided into subgroups after lesion: 1 day *(n =* 12), 3 days *(n =* 12), 8 days *(n =* 12), 30 days *(n =* 3) and 60 days *(n =* 3) -Group 2: Control non-operated *(n =* 12) -Group 3: Colchicine infusion *(n =* 3) -Group 4: TTX infusion *(n =* 3) -Group 5: Cardiotoxin injection *(n =* 3) -Group 6: Botulinum toxin injection *(n =* 3) -Group 7: Control-PBS injection *(n =* 6)	-Immunohistochemistry	-mRNAs encoding α1, β2, and γ2 subunits of GABA_A_ receptors were strongly downregulated in axotomized facial motoneurons as early as 3 days post-lesion and remained at low levels on post-lesion day 60. -mRNAs for GABA(B1B) and GABA(B2) receptor subunits were also downregulated by axotomy whereas those of GABA(B1A) remained unchanged. These changes in mRNA were accompanied by a decrease in GABA(B2) protein but not by a decrease in GABAB(1B) protein. -Colchicine reduced GABA_A_ α1 immunoreactivity and mRNA levels in the facial nucleus ipsilateral to the injected side. -TTX treatment decreased α1 GABA_A_ subunit expression in the lateral facial nucleus on day post-lesion 8. -Botulinum toxin had no effect at 1 week.	-Synaptic transmission of inhibitory inputs to facial motoneurons through GABA_A_ and GABA_B_ receptors is severely reduced by axotomy. -The loss of the GABA_A_ receptor α1 subunit was most likely the consequence of three phenomena: the loss of trophic factor transported from the periphery, a positive injury signal, and disruption in activity.
AMPAR NMDAR	Eleore et al., 2005 [136]	Male pigmented Long-Evans rats *(n =* 73)	-Left facial nerve section -TTX application for 8 days	-Group 1: Nerve axotomy *(n =* 55); divided into 1, 3, 8, 30 or 60 days after nerve injury -Group 2: TTX injection *(n =* 3) -Group 3: Control-sham operated *(n =* 11) -Group 4: Control-PBS injection *(n =* 4)	-In situ hybridization -Immunohistochemistry	-GLuR2-3 mRNAs were substantially reduced after facial nerve lesion; GLuR4 mRNA was downregulated less strongly. -mRNAs for NR1 and NR2A B and D subunits were lost from motoneurons following axotomy. Facial nerve axotomy resulted in a decrease in NR1 subunits in facial nuclei ipsilateral to the lesion. -VGLUT2 immunoreactivity in injured nuclei was lower 3 and 8 days after axotomy. -Catenin and pan-cadherin immunostaining was markedly decreased at the periphery of cell soma. -TTX caused facial palsy similar to that observed after facial nerve axotomy.	-Axotomy severely alters glutamatergic synaptic transmission in facial motoneurons at both post-synaptic and pre-synaptic levels. -The loss of AMPAR and NMDAR subunits is partly induced by a disruption in activity.
PSD-95 CAPON	Che et al., 2000 [137]	Male Sprague-Dawley rats *(n =* 32)	Left facial nerve transection	-Group 1: Nerve axotomy *(n =* 24); at postoperative days 1, 3, 5, 7, 14, 21, 28 and 35 *(n =* 3 for each period) -Group 2: Sham-operated *(n =* 8); at postoperative days 1, 3, 5, 7, 14, 21, 28 and 35 *(n =* 1 for each period)	-In situ hybridization -NADPH-d staining	-PSD-95 mRNA expression was decreased from postoperative day 1 to 7, gradually increased thereafter, and returned to constitutive levels at postoperative day 28. -CAPON mRNA was decreased from postoperative day 1 to 5 and increased thereafter, reaching constitutive levels at postoperative day 28. -Axotomized nerves started to reconnect to the muscles between postoperative day 7 and 14, and the number of WGA-positive neurons increased until postoperative day 35. -nNOS mRNA expression was increased from postoperative day 7 to just prior to the beginning of reinnervation of the muscle by the axotomized facial nerve.	-Recovery of PSD-95 and CAPON mRNA expression is correlated with reinnervation of muscles. -PSD-95 and CAPON are involved in synaptogenesis and recovery of synaptic function in motoneurons after axotomy.
nAChR α3 subunit	Senba et al., 1990 [138]	Male Sprague-Dawley rats *(n =* 19)	Left facial nerve transection	-Group 1: Nerve axotomy *(n =* 16); divided into postoperative survival times: 6 h *(n =* 2), 12 h *(n =* 3), 1 day *(n =* 4), 1 week *(n =* 3) and 2 weeks *(n =* 4) -Group 2: Control *(n =* 3)	-In situ hybridization	-α3 subunit mRNA expression on the operated side was decreased to ~1/3 of that on the control side 1 day after axotomy and completely disappeared 1 week after axotomy. -β2 subunit mRNA levels were enhanced in motoneurons on the operated side.	-The synthesis of α and β subunits of neuronal nAChRs is differentially regulated in axotomized motoneurons and the two subunits may play functionally different roles during the regeneration process.

Abbreviations: ChAT: choline acetyltransferase; KCC2: potassium sodium chloride cotransporter 2; FMN: facial motor nucleus; VAChT: vesicular acetylcholine transporter; GS: glycogen synthase; m2MAchR: m2 muscarinic acetylcholine receptor; OMgp: oligodendrocyte myelin glycoprotein; GABA: γ-aminobutyric acid; TTX: tetrodotoxin; PSD-95: post-synaptic density-95; CAPON: carboxy-terminal PDZ; nNOS: neuronal NOS.

**Table 3 antioxidants-12-01036-t003:** Biomolecules involved in the CNS after peripheral facial nerve injury.

Biomolecule/Process	Reference	Animal Model	Surgical Procedures	Experimental Design	Evaluations	Results	Conclusions
Autophagy	Hu et al., 2022 [39]	Male Sprague Dawley rats *(n =* 50)	Main trunk of the left facial nerve clamped for 60 s	-Group 1: Sham group *(n =* 10) -Group2: FNI group *(n =* 10) -Group 3: FNI + poloxamer *(n =* 10) -Group 4: FNI + bFGF *(n =* 10) -Group 5: FNI + P-bFGF *(n =* 10)	-Hematoxylin and eosin (H&E) or Masson’s trichrome staining -Immunofluorescence -Western blot	-P-bFGF improved functional recovery of early facial nerve injury. -P-bFGF upregulated the functional protein S100 in Schwann cells and boosted the remyelination of these cells. -P-bFGF treatment enhanced the fluorescence intensity of autophagy-related proteins (LC3B, f LC3B-II, Beclin1, and ATG5) and reduced the pro-apoptosis proteins cleaved caspase-3 and BAX.	-P-bFGF effectively promotes cell proliferation, myelination and functional recovery and also reduces apoptosis of nerve cells after FNI by activation of the PAK1 pathway in Schwann cells.
Autophagy	Gao et al., 2019 [185]	Male Sprague-Dawley rats *(n =* 80)	-Right extracranial facial nerve main trunk pressed for 50 s -CXCL12 injection at a dose of 4 μg/kg/d	Part I *(n =* 36); divided into 0, 1, 3, 7, 17 and 28 days -Group 1: Nerve injury -Group 2: Sham Part II *(n =* 44); divided into 3 and 28 days -Group1: Nerve injury -Group 2: Nerve injury + CXCL12	-H&E staining -Immunofluorescence -Western blot -Transmission electron microscopy	-Facial nerve injury enhanced the expression of CXCL12. CXCL12 significantly increased the migration of Schwann cells. -CXCL12 time-dependently increased autophagy of Schwann cells. -The autophagy inhibitor 3-MA significantly decreased CXCL12-induced expression of LC3II and increased expression of p62. -CXCL12 promoted Schwann cell migration through the PI3K/AKT/mTOR pathway. -CXCL12 promoted the recovery of facial nerve function and facilitated remyelination after facial nerve injury.	-CXCL12 has a therapeutic effect on facial nerve injury -CXCL12 acts through the PI3K/AKT/mTOR pathway to enhance autophagy and play a pivotal role in regulation of Schwann cell migration.
ROS	Chang et al., 2021 [187]	Female Wistar rats *(n =* NA)	Inoculation with 0.1 mL HHV7 virus solution	-Group 1: Normal control -Group 2: HHV7 infection -Group3: HHV7 infection + shNC -Group4: HHV7 infection + shCoxi42	-Luxol Fast Blue staining -Immunofluorescence -Western blot -Flow cytometry -Phen Green SK staining -TUNEL assay	-Increased expression of Cox4i2 in Schwann cells infected with HHV7 promoted the production of ROS, and knockdown of Cox4i2 expression in HHV7-infected Schwann cells induced a relative decrease in ROS levels. -Increased expression of Cox4i2 led to an increase in ROS production in HHV7-infected Schwann cells that subsequently induced ferroptosis. Conversely, ferroptosis was inhibited by knock down of Cox4i2 in HHV7-infected Schwann cells.	-This study revealed a new mechanism of ROS-induced and Cox4i2-mediated apoptosis and ferroptosis in HHV7-infected Schwann cells.
IL-10	Villacampa et al., 2015 [26]	GFAP-IL-10Tg *(n =* 66) Wild-type *(n =* 61)	Right facial nerve 1 mm resection	-Group 1: GFAP-IL-10Tg -Group 2: GFAP-IL-10Tg + Rt facial nerve axotomy -Group 3: Wild type -Group 4: Wild type + right facial nerve axotomy	-RT-PCR -Toluidine blue staining -Immunohistochemistry	-Greater CD3 positive lymphocyte infiltration was observed in the axotomized facial nerve of GFAP-IL-10Tg mice. -Astrocyte-targeted IL-10 production showed a strong beneficial effect on neuronal survival. -FMN constitutively express IL-10R; after facial nerve axotomy, IL-10R expression was lower but was maintained at all time-points. -Expression of CD39, an ectonucleotidase highly expressed in M2 macrophages, was increased in activated microglia from wild-type and GFAP-IL-10Tg animals after facial nerve axotomy. -The number of microglial cells significantly increased at 3 and 7 dpi on the lesioned facial nerve.	-IL-10 production within the CNS can lead to significant modifications in the pattern of microglial activation and T-cell infiltration and may exert a beneficial effect on the outcome of peripheral nerve injury.
Calcium	Mattsson et al., 1999 [188]	Male Sprague-Dawley rats *(n =* 37)	-Right facial nerve transection -Nimodipine administration from 3 days before the operation until death	-Group 1: Nerve axotomy *(n =* 14) -Group 2: Nerve axotomy + nimodipine *(n =* 16) -Group 3: Sham-operated animal *(n =* 7)	-Immunocytochemistry	-Nimodipine, a well-known antagonist of calcium influx, was shown to be neuroprotective after various lesions in the CNS. -OX42 immunoreactivity ipsilateral to the nerve injury increased from 2 to 7 days post-injury and remained at this level up to 28 days post-injury, in Groups 1 and 2. -ED1 immunoreactivity was increased ipsilateral to the nerve lesion from 2 to 28 days post-injury in Groups 1 and 2.	-Nimodipine, a calcium channel blocker that enhances blood flow and reduces ischemia, significantly improved neuronal survival for at least 1 month after oral administration in rats with intracranial transection of the facial nerve. -Nimodipine may have potential as a neuroprotective agent for various types of nerve injury.

Abbreviations: bFGF: basic fibroblast growth factor; FNI: facial nerve injury; PAK1: P21-activated kinase 1; CXCL12: C-X-C Motif Chemokine Ligand 12; PI3K: phosphoinositide 3-kinases; mTOR: mammalian target of rapamycin; Cox4i2: complex IV subunit 4 isoform 2; HHV7: herpesvirus 7; ROS: reactive oxygen species; CD3: cluster of differentiation 3; GFAP-IL: glial fibrillary acidic protein- interleukin; CNS: central nervous system; OX42: oxycocin-42.

**Table 4 antioxidants-12-01036-t004:** Facial nerve injury can result in a variety of changes in the CNS.

Experimental Focus	Reference	Animal Model	Surgical Procedures	Experimental Design	Evaluations	Results	Conclusions
Cortex	Lee et al., 2016 [38]	Human *(n =* 37)	-	-Group 1: Recovered palsy *(n =* 17; male, 8, female, 9) -Group 2: Control *(n =* 20; female, 7, male, 13)	-Siemens Symphony 1.5 T MRI whole-body scanning -3D anatomical imaging -Task-state fMRI	-Cortical reorganization persisted in patients recovered from Bell’s palsy, or the functional status of the brain had not returned to the normal condition before the disease. -Activation significantly increased in the posterior cingulate cortex (PCC), primary somatosensory cortex (SI), primary motor cortex (MI) and cingulate motor area (CMA), and decreased in the parahippocampal gyrus during finger movements. -Signals decreased in the SI, PCC, precuneus, and culmen during lip pursing movements. -Cerebral blood flow in the facial motor area of the brain in patients recovered from Bell’s palsy was reduced, whereas that in hand motor areas was enhanced.	-Regions showing changes in activation between the two groups included the motor association cortex and cerebellum. -All of these changes in the cortex might be relevant to differences in the functional status of the brain.
Discharge properties	Shi et al., 2016 [10]	Female Wistar rats *(n =* 8)	-Transection of the right trunk of the facial nerve -Implantation of a 4 × 4 electrode arrays into the brainstem on the right	-Group1: Right facial nerve transection *(n =* 8) -Group 2: Non-injured side *(n =* 8)	-Toluidine blue staining -Transmission electron microscopy	-Nerve degeneration, manifested as disorderly distributed smaller fibers with demyelination and swelling of organelles, was detected in the injured group compared with Group 2. -The sustained spike pattern of neuron A changed to a phasic pattern and the firing rate of neuron B decreased compared with its original firing rate. -The mean frequency, coefficient of variation median ISI and modal ISI were significantly changed in both neuron A and neuron B during execution of movements following neurotmesis.	-Neurotmesis attenuated nerve firing rates, and changed firing patterns throughout the duration of movements, all of which may provide a theoretical basis for observed facial palsy, synkinesia, and prosopospasm.
Nucleus	Jemec et al., 2000 [29]	Human *(n =* 21)	-	-Group: Patients with CFP *(n =* 21, male, 7, female, 14); includes those with CFP as their sole symptom *(n =* 15) and with syndromes that can include CFP *(n =* 5)	-MRI scan	-Five (24%) of the abnormal scans showed a lesion in the area of the facial nucleus in the pons; in two, the nucleus was completely absent, and in the other three, T1/T2 weighting in the area was abnormal. -Three (14%) of the abnormal scans showed other abnormalities, including a prominent circle of Willis, partial agenesis of the corpus callosum, and cerebellar hypoplasia.	-Developmental abnormalities of the facial nucleus itself constitute an important, and previously ignored, cause of monosymptomatic unilateral CFP.
Motor neurons	Nakao et al., 1992 [205]	Rabbits *(n =* 15)	Facial nerve crush	-Group 1: Control *(n =* 10) -Group 2: Nerve crush *(n =* 5)	-Immunohistochemistry	-After recovery from facial nerve paralysis, labeled neurons innervating the zygomatic muscle were located not only in the ventromedial portion but also partially in the dorsomedial portion. -Multipolar neurons of varying size were labeled bilaterally in the reticular formation from the pons to the medulla. These neurons contained HRP granules that were brown in color but paler than those in the facial nucleus.	-Labeled premotor neurons may have a role in muscle movements after recovery from facial nerve palsy.
Cell body	Totoki et al., 1980 [206]	Male Japanese monkeys *(n =* 4)	Block of the neural canal of the facial nerve by insertion of a 22-gauge needle	-Group 1: Control *(n =* 1) -Group 2: Nerve block *(n =* 3)	-Toluidine blue staining	-Nerve cells of the facial nucleus were circular, with their nuclei shifted to one side of the cell, 4 days after nerve block. -Nissl staining was dispersed and the size of Nissl granules was decreased in neurons. -Nissl granules were still small and nuclei were positioned towards one side of the cell 2 months after nerve block. -Nerve cells in the facial nucleus were nearly normal in the blocked side 7 months after nerve block, but recovery was incomplete and small Nissl granules were present in some cells.	-Nerve function recovered in 2 months, but 7 months were required for nerve cells in the facial nerve nucleus to recover completely. -A mechanism to account for the complete remission in patients ~6 months after nerve block is suggested.

Abbreviations: PCC: posterior cingulate cortex; SI: primary somatosensory cortex; MI: primary motor cortex; CMA: cingulate motor area; ISI: inter spike intervals; fMRI: functional magnetic resonance imaging; CFP: Congenital facial palsy; HRP: horseradish peroxidase.

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
