# Peer review of "Central Facial Nervous System Biomolecules Involved in Peripheral Facial Nerve Injury Responses and Potential Therapeutic Strategies"

_antioxidants, 2023, doi:10.3390/antiox12051036_

Round 1

Reviewer 1 Report

This is a very well written and comprehensive review of peripheral facial nerve injury and its treatment options.
The only minimal "criticism" would be that there are slightly redundant passages in some paragraphs.
For the sake of consistency, it would be good if the number of animals was given in all tables (if the study itself does not specify this, one can write n= NA). 

I find the title a bit long and bulky.
Of course, illustrations would also be nice, but with so much information it is not easy to summarise it compactly, e.g. in a graphic abstract.

Author Response

Response to Reviewer #1 Comments

This is a very well written and comprehensive review of peripheral facial nerve injury and its treatment options.

The only minimal "criticism" would be that there are slightly redundant passages in some paragraphs.

For the sake of consistency, it would be good if the number of animals was given in all tables (if the study itself does not specify this, one can write n= NA).

Response:

We appreciate the constructive comments made by the reviewer. We have revised the manuscript according to the suggestions. For consistency, the number of animals is shown in all tables. If not specified in the paper, the number is indicated as (n=NA).

I find the title a bit long and bulky.

Of course, illustrations would also be nice, but with so much information it is not easy to summarise it compactly, e.g. in a graphic abstract.

Response:

We have added a schematic diagram of the molecules altered in the central nervous system following facial nerve injury.

Reviewer 2 Report

This is a review manuscript discusses the changes with a focus on facial nerve damage. The authors reviewed 30 experimental studies to identify the biomolecules involved in these changes and potential treatment strategies for facial nerve injury. By understanding the biomolecules involved in the CNS changes, the authors hope to identify factors important for functional recovery from facial nerve damage and develop more effective treatment strategies for peripheral facial palsy. Basically, I very much agree with the contributions of the authors in this regard, especially the systematic arrangement of relevant materials in detail. However, I still have some comments that I would like to propose to the authors for their reference.

1. Looking at the entire manuscript, peripheral nerve injury and regeneration seem to be the focus of the authors' discussion, especially nerve regeneration is a hot topic in recent research. However, we all know that in addition to neuron, Schwann cells and macrophages are also important participants in the above-mentioned procedures, but the authors describe the role of macrophage (or related molecular changes) quite rarely? I think this may be a part that needs to be strengthened.

2. After reading the author's text, I still don't quite understand the difference between peripheral nerve injury and peripheral "facial" nerve injury? Maybe the authors can further strengthen the uniqueness of facial nerve injury.

3. In fact, the authors have described the changes of each related molecule in detail, but I think it seems that we can go a little further, for example, what possible clinical treatments may affect the changes of these molecules? Or how to regulate these molecules for management of peripheral facial nerve injury. I think the authors would add a discussion of this aspect at least in the final section to complete the review.

4. If possible, I would suggest that the authors provide another scheme figure, and mark the changed molecules and their possible physiological roles in nerve regeneration, which will be easier for readers to understand than the table content.

Author Response

Response to Reviewer #2 Comments

This is a review manuscript discusses the changes with a focus on facial nerve damage. The authors reviewed 30 experimental studies to identify the biomolecules involved in these changes and potential treatment strategies for facial nerve injury. By understanding the biomolecules involved in the CNS changes, the authors hope to identify factors important for functional recovery from facial nerve damage and develop more effective treatment strategies for peripheral facial palsy. Basically, I very much agree with the contributions of the authors in this regard, especially the systematic arrangement of relevant materials in detail. However, I still have some comments that I would like to propose to the authors for their reference.

  1. Looking at the entire manuscript, peripheral nerve injury and regeneration seem to be the focus of the authors' discussion, especially nerve regeneration is a hot topic in recent research. However, we all know that in addition to neuron, Schwann cells and macrophages are also important participants in the above-mentioned procedures, but the authors describe the role of macrophage (or related molecular changes) quite rarely? I think this may be a part that needs to be strengthened.

Response:

We appreciate the constructive comments made by the reviewer. We have revised the manuscript according to the suggestions made. Following the reviewer's advice, we have added a section on future perspectives to the Discussion section of the manuscript. The text has been changed to the following (Lines 697-711):

One critical process in maintaining cellular and tissue homeostasis is autophagy, which removes damaged organelles, pathological proteins, and dysfunctional macromolecules from within cells. Macrophages are crucial players in nerve injury and regeneration. After nerve injury, macrophages trigger an inflammatory response and help to clear and regenerate damaged tissue [217]. Moreover, the coordinated actions of the M1 and M2 subtypes can establish a favorable microenvironment for the release of cytokines that aid in the repair of damaged tissue [218]. As a result, macrophages play a critical role in both nerve damage and regeneration. After facial nerve injury, activation of autophagy is beneficial for reducing scarring and promoting myelination, which facilitates regeneration and motor recovery [190]. ROS levels also increase significantly after peripheral nerve injury, leading to cellular damage through oxidative stress [219]. Thus, inhibiting ROS production could mitigate oxidative damage induced by facial nerve injury. Moreover, IL-10 plays a significant role in regulating inflammatory and immune responses following facial nerve injury [204]. Inhibition of IL-10 reduces facial nerve survival, whereas overproduction of IL-10 protects against facial nerve injury by preventing neuronal cell death.

  1. After reading the author's text, I still don't quite understand the difference between peripheral nerve injury and peripheral "facial" nerve injury? Maybe the authors can further strengthen the uniqueness of facial nerve injury.

Response:

Following the advice of the reviewers, we unified the text on facial nerve injury to avoid confusion about peripheral and facial nerve injuries.

  1. In fact, the authors have described the changes of each related molecule in detail, but I think it seems that we can go a little further, for example, what possible clinical treatments may affect the changes of these molecules? Or how to regulate these molecules for management of peripheral facial nerve injury. I think the authors would add a discussion of this aspect at least in the final section to complete the review.

Response:

We added to the Discussion a statement regarding the potential impacts of changes in these molecules on clinical treatment based on our review. The text has been changed to the following (Lines 765-774):

The present review of the biomolecules and processes that change in the CNS after facial nerve injury provides insight into how peripheral facial nerve injury affects the CNS and suggests strategies for harnessing this information to promote recovery from injury. To treat patients with facial nerve damage, it is necessary to control the signal transmission system of the central facial nerve and/or to discover and develop a control agent or a nerve regeneration agent. Studying changes in the biomolecules involved in facial nerve injury should enable identification of the factors that play important roles in functional recovery of facial nerve damage, which can then be exploited in clinical treatment. The results summarized in this review have implications for various treatments, including facial nerve damage surgery and drug treatment for facial nerve injury.

  1. If possible, I would suggest that the authors provide another scheme figure, and mark the changed molecules and their possible physiological roles in nerve regeneration, which will be easier for readers to understand than the table content.

Response:

We have added a schematic diagram of the molecules altered in the central nervous system following facial nerve injury.

Reviewer 3 Report

In this manuscript the authors reviewed the information about the biomolecules involved in peripheral facial nerve damage so as to gain insight into the mechanisms and limitations of targeting the CNS after such damage and identify potential facial nerve treatment strategies

The idea of this study is principally interesting, and this review is well-written; unfortunately, this manuscript needs real some improvements and corrections before publishing may be possible.

General points:

For better readability, please add 1-3 Figures to your manuscript.

Please add a list of abbreviations before References section to your manuscript.

Please add a Future perspectives section to your manuscript.

Special points:

Keywords: please add to keywords: treatment; peripheral facial nerve injury.

Introduction

Lines 34-49: please add multiple references at the end of each of these sentences.

Lines 69-93: please add multiple references at the end of each of these sentences.

2. Changes in the CNS after peripheral nerve injury

Lines 109-119: please add multiple references at the end of each these of sentences.

Lines 133-153: please add multiple references at the end of each these of sentences.

3. Biomolecules increased in the CNS after peripheral facial nerve injury

Lines 169-199: please add multiple references at the end of each of these sentences.

Table 1: please say: Abbreviations: GPR37-IR: G protein-coupled receptor; SHARPIN: shank-associated RH domain-interacting protein; HSV-1: herpes simplex virus-1; NOS: nitric oxide synthase; FMN: facial motor nucleus; NADPH-diaphorase: nicotinamide-adenine-dinucleotide phosphate-diaphorase; iNOS: inducible NOS; NO: nitric oxide; VIP: vasoactive intestinal peptide ; SP: substance P; CD3: cluster of differentiation 3; IL-6: interleukin-6; FGF-2: fibroblast growth factor-2; GFAP: glial fibrillary acidic protein; Shh: sonic hedgehog; Smo: smoothened; CGRP: calcitonin gene-related peptide; GAP-43: growth-associated protein-43; GDNF: glial cell line-derived neurotrophic factor; CNS: central nervous system; BDNF: brain-derived neurotrophic factor; TrkB: tropomyosin receptor kinase B

Lines 245-258: please add multiple references at the end of each of these sentences.

Lines 270-281: please add multiple references at the end of each of these sentences.

Lines 288-311: please add multiple references at the end of each of these sentences.

Lines 328-332: please add multiple references at the end of this sentence.

Lines 341-343: please add multiple references at the end of each of these sentences.

Lines 358-364: please add multiple references at the end of each of these sentences.

Lines 393-407: please add multiple references at the end of each of these sentences.

4. Biomolecules decreased in the CNS after peripheral facial nerve injury

Lines 423-425:  please add multiple references at the end of this sentence.

Lines 427-428: please add multiple references at the end of this sentence.

Table 2: please correct in the same way with Table 1. See my proposal above.

Lines 445-446: please add multiple references at the end of this sentence.

Lines 464-480: please add multiple references at the end of each of these sentences.

Lines 491-493: please add multiple references at the end of this sentence.

Lines 497-499: please add multiple references at the end of this sentence.

Lines 510-520: please add multiple references at the end of each of these sentences.

Lines 530-541: please add multiple references at the end of each of these sentences.

Lines 549-563: please add multiple references at the end of each of these sentences.

Lines 567-569: please add multiple references at the end of this sentence.

Line 572: please add multiple references at the end of this sentence.

Lines 582-590: please add multiple references at the end of each of these sentences.

5. Central facial nerve biomolecules and processes involved in peripheral facial nerve damage

Lines 601-602: please add multiple references at the end of this sentence.

Table 3: please correct in the same way with Table 1. See my proposal above.

Lines 624-637: please add multiple references at the end of each of these sentences.

Lines 648-657: please add multiple references at the end of each of these sentences.

Line 661: please add multiple references at the end of this sentence.

Lines 668-671: please add multiple references at the end of each of these sentences.

Lines 676-687: please add multiple references at the end of each of these sentences.

Table 4: please correct in the same way with Table 1. See my proposal above.

Discussion

Lines 705-752: please add multiple references at the end of each of these sentences.

Author Response

Response to Reviewer #3 Comments

In this manuscript the authors reviewed the information about the biomolecules involved in peripheral facial nerve damage so as to gain insight into the mechanisms and limitations of targeting the CNS after such damage and identify potential facial nerve treatment strategies

The idea of this study is principally interesting, and this review is well-written; unfortunately, this manuscript needs real some improvements and corrections before publishing may be possible.

General points:

For better readability, please add 1-3 Figures to your manuscript.

Response: We appreciate the constructive comments made by reviewers and giving us the opportunity for revision. We have added figures to the manuscript for readability, following the reviewer's advice.

Please add a list of abbreviations before References section to your manuscript.

Response: We have added an abbreviation list before the References section.

Abbreviations

GPR37-IR: G protein-coupled receptor 37; SHARPIN: shank-associated RH domain-interacting protein; HSV-1: herpes simplex virus-1; NOS: nitric oxide synthase; FMN: facial motor nucleus; NADPH-diaphorase: nicotinamide-adenine-dinucleotide phosphate-diaphorase; iNOS: inducible NOS; NO: nitric oxide; VIP: vasoactive intestinal peptide ; SP: substance P; CD3: cluster of differentiation 3; IL-6: interleukin-6; FGF-2: fibroblast growth factor-2; GFAP: glial fibrillary acidic protein; Shh: sonic hedgehog; Smo: smoothened; CGRP: calcitonin gene-related peptide; GAP-43: growth-associated protein-43; GDNF: glial cell line-derived neurotrophic factor; CNS: central nervous system; BDNF: brain-derived neurotrophic factor; TrkB: tropomyosin receptor kinase B; ChAT: choline acetyltransferase; KCC2: potassium sodium chloride cotransporter 2; FMN: facial motor nucleus; VAChT: vesicular acetylcholine transporter; GS: glycogen synthase; m2MAchR: m2 muscarinic acetylcholine receptor; OMgp: oligodendrocyte myelin glycoprotein; GABA: γ-aminobutyric acid; TTX: tetrodotoxin; PSD-95: post-synaptic density-95; CAPON: carboxy-terminal PDZ; nNOS: neuronal NO; bFGF: basic fibroblast growth factor; FNI: facial nerve injury; PAK1: P21-activated kinase 1; CXCL12: C-X-C Motif Chemokine Ligand 12; PI3K: phosphoinositide 3-kinases; mTOR: mammalian target of rapamycin; Cox4i2: complex IV subunit 4 isoform 2; HHV7: herpesvirus 7; ROS: reactive oxygen species; CD3: cluster of differentiation 3; GFAP-IL: glial fibrillary acidic protein- interleukin; CNS: central nervous system; OX42: oxycocin-42; PCC: posterior cingulate cortex; SI: primary somatosensory cortex; MI: primary motor cortex; CMA: cingulate motor area; ISI: inter spike intervals; fMRI: functional magnetic resonance imaging; CFP: Congenital facial palsy; HRP: horseradish peroxidase

Please add a Future perspectives section to your manuscript.

Response: Following the reviewer's advice, we have added a section on future perspectives to the discussion section of the manuscript. The text has been changed to the following (Lines 765-774):

The present review of the biomolecules and processes that change in the CNS after facial nerve injury provides insight into how peripheral facial nerve injury affects the CNS and suggests strategies for harnessing this information to promote recovery from injury. To treat patients with facial nerve damage, it is necessary to control the signal transmission system of the central facial nerve and/or to discover and develop a control agent or a nerve regeneration agent. Studying changes in the biomolecules involved in facial nerve injury should enable identification of the factors that play important roles in functional recovery of facial nerve damage, which can then be exploited in clinical treatment. The results summarized in this review have implications for various treatments, including facial nerve damage surgery and drug treatment for facial nerve injury.

Special points:

Keywords: please add to keywords: treatment; peripheral facial nerve injury.

Response: We added “treatment” and “peripheral facial nerve injury” to the keywords.

Keywords: peripheral facial nerve injury; facial paralysis treatment; central nervous system; biomolecules; facial motor neurons

Introduction

Lines 34-49: please add multiple references at the end of each of these sentences.

Response: As per the reviewer’s suggestion, references have been added at the end of each sentence (Lines 34-49).

The facial nerve, also known as the seventh cranial nerve (CN VII), is a mixed nerve composed of motor, sensory, and parasympathetic nerve fibers [1]. The facial nerve not only carries nerve impulses that control the muscles responsible for facial expression and eye blinking, they also regulate tear and salivary gland secretion, tongue movement, and sensation of the soft palate [2]. Because the facial nerve is longer than other cranial nerves and passes through a narrow canal as it exits the temporal bone, it is more vulnerable to damage caused by middle ear and temporal bone surgery, trauma, and infection [3]. Central facial palsy, a condition in which damage to the brain by various factors, such as infection, tumor growth or brain tumor, results in the appearance of symptoms on the contralateral side of the face and a reduction in facial muscle function [4]. In peripheral facial paralysis, functional abnormalities of the facial nerve itself appear, and symptoms occur on the same side (ipsilateral) as the lesion, resulting in reduced facial muscle function [5].

Damage to the peripheral nervous system can result in a loss of function and a reduced quality of life owing to impaired motor function [6]. One manifestation of this functional loss is peripheral facial nerve palsy, which causes paralysis of the facial expression muscles on one side [7].

Lines 69-93: please add multiple references at the end of each of these sentences.

 Response: We have added references at the end of each sentence (Lines 69-93).

Nerve regeneration in the peripheral nervous system can lead to functional recovery, but the extent of recovery can vary depending on the severity and location of the nerve injury [6]. Whereas peripheral nerves have some capacity for neural plasticity and regeneration, the process is often slow and incomplete, and only partial functional recovery may be achieved [17]. Injuries that are closer to the spinal cord or involve larger nerve trunks may have a more significant impact on functional recovery because of the greater complexity and length of the nerve pathway [18]. Damaged peripheral neurons regenerate axons over a long period of time at a very slow rate (~1 mm per day). If the rate of regeneration is slow, it may take months or even years for the reinnervation of functional motor units or sensory organs to occur [19]. Following peripheral nerve injury, a number of factors involved in the process of axon regeneration show immediate up- or downregulated expression [20]. Such changes in the expression of regeneration-related genes, together with the degeneration and removal of myelin and axons, are the main features of the Wallerian degeneration process [21].

Following peripheral nerve injury, damage signals cause changes in the expression of genes encoding proteins that activate the regenerative response in neurons [22]. This leads to changes in the synthesis of molecules, including neurotransmitters, cytoskeletal proteins and growth-associated proteins, in the central nervous system (CNS) [23]. In adult rats, transection of the facial nerve results in motor neuron damage, and after ischemic peripheral facial paralysis, the expression of genes such as c-Jun as well as growth-associated proteins are altered in facial nerve nuclei [24]. These alterations result in changes in various other biomolecules, including cytoskeletal molecules, metabolic enzymes, neuropeptides, and cytokines. Additionally, the expression of choline acetyltransferase, which is related to motor neuron function, is reported to be reduced in the motor nucleus of the brainstem [25].

  1. Changes in the CNS after peripheral nerve injury

Lines 109-119: please add multiple references at the end of each these of sentences.

 Response: We have added references at the end of each sentence (Lines 109-118).

Peripheral facial nerve injury can cause partial or complete loss of motor, sensory, and autonomic functions in affected areas of the body, reflecting the interruption of nerve fibers and the resulting disruption in axonal continuity and degeneration [7]. After facial nerve axonal injury, the range and number of synaptic terminals of motor neurons in the facial nucleus are reduced [26]. The degeneration of motor neurons is often accompanied by the activation of microglia located in the facial nucleus [27]. The facial motor control circuitry in the CNS is able to recover to some extent after peripheral nerve damage, as evidenced by the preservation of the overall structure of the facial nucleus and the formation of reticular formations from the pons to the medulla by various types of multipolar neurons after recovery from facial nerve palsy [28].

Lines 133-153: please add multiple references at the end of each these of sentences.

Response: We have added references at the end of each sentence (Lines 132-153).

Neuronal plasticity is a common characteristic of the nervous system that enables neurons to adapt and modify their structure and function in response to various environmental signals, learning processes, injury, and disease [32]. Several studies have described functional plasticity in the context of various pathologies, including brain lesions and peripheral nerve transection [33,34]. This plasticity helps to restore damaged peripheral nerves by establishing an effective connection between the nervous system and the target tissue and by regulating the functional remodeling of the CNS [35]. Reconstruction in the spinal cord, brainstem, thalamus, and cortex following peripheral nerve damage has been confirmed using brain imaging techniques [36]. Cortical reorganization or cortical plasticity, which refers to the brain’s ability to change its neural connections and function in response to new experiences or changes in the environment, is a particularly evident phenomenon in the cortex—the outer layer of the brain responsible for higher cognitive functions such as perception, language, and memory [37]. Cortical plasticity plays a crucial role in learning and memory, recovery from brain injury, and adaptation to changes in sensory input [38]. Cortical reorganization has been observed in recovering Bell’s palsy patients by monitoring brain activity using functional magnetic resonance imaging (fMRI) during finger and facial movements [39]. During the Bell’s palsy recovery period, there is an increase in connectivity between the ipsilateral and contralateral anterior cingulate cortex and strengthening of the functional relationship between the unaffected anterior cingulate cortex and the sensorimotor area that contributes to adjustments in abnormal facial movements [40]. This improved connectivity of the ipsilateral and contralateral anterior cingulate cortex is the result of monitoring and compensatory functions [41].

  1. Biomolecules increased in the CNS after peripheral facial nerve injury

Lines 169-199: please add multiple references at the end of each of these sentences.

 Response: We have added references at the end of each sentence (Lines 169-201).

PS is a precursor protein that is cleaved into four smaller proteins, saposins A, B, C, and D [45]. These saposins function as coenzymes with sphingolipid activator proteins and are involved in the breakdown of sphingolipids, which are important components of cell membranes in the nervous system [46]. PS has also been shown to exert neuroprotective and neurotrophic effects on neurons and glial cells, and is expressed after nerve damage as part of the process of nerve regeneration and repair [47]. PS is transported to lysosomes, where it undergoes proteolytic processing into the four saposins, which are required for normal hydrolysis of sphingolipids [48]. PS levels were found to be significantly increased in the facial nerve nucleus after facial nerve transection, suggesting the activation of various neurotrophic activities in facial nerve cells [49]. The addition of PS to collagen-filled nerve guides after sciatic nerve transection in guinea pigs was found to promote increased peripheral nerve regeneration within the guide [50]. PS administration was additionally shown to help ameliorate atrophy of spinal anterior horn and dorsal root ganglion neurons [51]. The expression of PS mRNA was found to be increased in a rat model of focal cerebral ischemia and cortical injury, suggesting its potential role in regulating cerebral nerve regeneration [52]. PS has been shown to have protective effects in the nervous system and to play a role in the activation of G proteins. For example, the orphan G protein-coupled receptors (GPCRs), GPR37 and GPR37L1, which are expressed in neurons and glial cells in the nervous system, are thought to be involved in mediating the effects of PS [53]. Both GPR37 and GPR37L1 are known to stimulate self-binding of prosaptide, which in turn activates signaling pathways that promote endocytosis in the nervous system [54]. In addition, small interfering RNA (siRNA)-mediated knockdown of endogenous astrocyte GPR37 and GPR37L1 was reported to attenuate the protective effects of prosaptide and PS on astrocytes [55]. GPR37 and GPR37L1 were found to be increased in microglia and astrocytes in the facial nuclei of mice following facial nerve transection [56]. Whereas GPR37 mainly acts in neurons, GPR37L1 is predominantly expressed in microglia or astrocytes. Increased PS in damaged neurons produces neurotrophic factors through GPR37L1, which is involved in nerve recovery [56]. Hippocampal and cortical neurons show increased immunoreactivity and expression of PS mRNA following kainic acid-induced excitotoxicity, and increased PS levels were shown to improve neuronal survival by promoting the delivery of lysosomal enzymes to damaged neurons after injury [57] (Table 1).

Table 1: please say: Abbreviations: GPR37-IR: G protein-coupled receptor; SHARPIN: shank-associated RH domain-interacting protein; HSV-1: herpes simplex virus-1; NOS: nitric oxide synthase; FMN: facial motor nucleus; NADPH-diaphorase: nicotinamide-adenine-dinucleotide phosphate-diaphorase; iNOS: inducible NOS; NO: nitric oxide; VIP: vasoactive intestinal peptide ; SP: substance P; CD3: cluster of differentiation 3; IL-6: interleukin-6; FGF-2: fibroblast growth factor-2; GFAP: glial fibrillary acidic protein; Shh: sonic hedgehog; Smo: smoothened; CGRP: calcitonin gene-related peptide; GAP-43: growth-associated protein-43; GDNF: glial cell line-derived neurotrophic factor; CNS: central nervous system; BDNF: brain-derived neurotrophic factor; TrkB: tropomyosin receptor kinase B

 Response: Following the reviewer's advice, we have defined the abbreviations in Tables 1-4 at the bottom of the table. We have also added an abbreviation list before the References section.

Lines 245-258: please add multiple references at the end of each of these sentences.

Response: We have added references at the end of each sentence (Lines 244-259).

NO plays a critical role in both non-specific and immunological host defense and has antibacterial effects through cytotoxic or cytostatic actions against various pathogens [71]. For example, NO produced by iNOS has beneficial effects on host defense mechanisms against bacteria [72]. However, in the case of HSV-1-associated facial nerve palsy, NO contributes to the pathogenesis of neuroviral infections and neurodegeneration [73]. During facial paralysis caused by facial nerve compression, NOS and NADPH-diaphorase activity are increased in facial motor neurons [74]. The resulting increase in NADPH-diaphorase activity contributes to the recovery of facial function by promoting axon regeneration [75]. In HSV-1-infected mice, iNOS-induced NO is overproduced in neurons and HSV-1 increases apoptosis in the brainstem of mice [76]. Facial nerve damage caused by nerve compression leads to increased NOS activity and NO production in facial motor neurons and surrounding tissues [77]. This effect is attributable to the extensive activation of NMDA receptors, which mediate the effects of the excitatory neurotransmitter glutamate and have been implicated in neuronal cell death [66]. In the context of facial nerve damage, release of NO by activation of NMDA receptors increases blood flow to the injured area, inducing local vasodilation [78].

Lines 270-281: please add multiple references at the end of each of these sentences.

Response: We have added references at the end of each sentence (Lines 266-278).

In addition, VIP can help resolve acute inflammatory processes and may contribute to the prevention of chronic inflammation [81]. Changes in gut flora and biodiversity, as well as weight loss, have been observed in VIP-deficient mice, which show increased susceptibility to intestinal inflammation and inflammatory bowel disease [82]. In a spinal cord injury model, VIP was shown to inhibit the induction of TNFα and interleukin (IL)-6 in microglia, reducing neuronal cell loss around the lesion site [83]. Substance P (SP) is a tachykinin neuropeptide that acts as a neurotransmitter and neuromodulator in the CNS [84]. After peripheral facial nerve axotomy, SP and VIP expression are strongly increased in the facial nucleus, where the resulting early T cell recruitment is accompanied by increased levels of the proinflammatory cytokine, IL-6 [85]. In the absence of IL-6 (IL-6-deficient mice), lymphocyte recruitment and axonal regeneration are reduced and there is a decrease in CD3-positive T-lymphocyte recruitment and early microglia activation [86].

3.5. Fibroblast growth factor-2 and glial fibrillary acidic protein

Fibroblast growth factor-2 (FGF-2), a member of the fibroblast growth factor family, is known to have a variety of biological functions, including promoting cell proliferation, survival and angiogenesis [87]. It is also known to play a critical role in tissue repair and wound healing, particularly in the skin and bone. In addition, it is involved in embryonic development and organogenesis, and has been shown to play a role in cancer progression [88].

Lines 288-311: please add multiple references at the end of each of these sentences.

Response: We have added references at the end of each sentence (Lines 289-314).

FGF-2 mRNA and protein are widely expressed in the brain, with the highest levels observed in astrocytes [91]. After injury or insult to the brain, FGF-2 expression is upregulated primarily in astrocytes, and its release from astrocytes plays an important role in promoting the survival and proliferation of neural progenitor cells [92]. A treatment strategy using Schwann cells overexpressing FGF-2, alone or in combination with passive stimulation, after facial peripheral nerve transection was shown to induce transient lateral branching by supporting axon regeneration, but did not significantly improve functional recovery after facial nerve injury [93]. FGF-2 isoforms are upregulated in spinal cord neurons and sciatic nerves after peripheral nerve lesions [94]. Unilateral compression or transection of the lingual nerve was reported to increase the number of FGF-2-immunoreactive neurons and glia, and increase the amount of FGF-2 present in reactive astrocytes of the lingual nerve nucleus [95].

Activated astrocytes, characterized by their expression of glial fibrillary acidic protein (GFAP), accumulate around nerve cells, where the FGF-2 they produce acts as a neuroprotective factor [96]. Axotomy increases the number of GFAP-positive astrocytes in the facial nucleus and enhances their nuclear expression of FGF-2 [97]. The resulting increase in FGF-2 in the cytoplasm of reactive astrocytes leads to enhanced secretion of FGF-2, which acts in a paracrine and autocrine manner to provide trophic support to the facial nucleus, thereby preventing Bell’s palsy [98]. Astrocyte activation in the CNS after peripheral nerve injury contributes to nerve regeneration by maintaining immune homeostasis [99], reflecting the critical role of astrocytes in restoring the blood-brain barrier, providing neuroprotection, and limiting the proliferation of inflammatory cells [100]. This supportive function of astrocytes is considered an important factor in the survival of damaged neurons in the CNS and maintenance of synaptic plasticity and neurotransmitter release [101]. The expression of GFAP is regulated by various hormones, cytokines, and growth factors [102].

Lines 328-332: please add multiple references at the end of this sentence.

Response: We have added references at the end of each sentence (Lines 324-331).

During neocortex development, Shh signaling regulates intermediate progenitors to maintain neuron proliferation, survival, and differentiation in the neocortex [107]. Shh protein expression and Smo mRNA levels are upregulated in facial motor neurons after facial nerve axotomy in adult rats, and adenoviral-mediated overexpression of Shh is associated with the survival of axotomized motor neurons [105]. Shh has been shown to regulate stem cell proliferation in the adult rat hippocampus, and overexpression of Shh in the forebrain improves cognitive and motor impairment [108,109].

3.7. Calcitonin gene-related peptide and growth-associated protein-43

The molecular response of damaged motor neurons following facial nerve injury involves the upregulation of early genes followed by the expression of neuromodulatory and regeneration-related genes [110].

Lines 341-343: please add multiple references at the end of each of these sentences.

Response: We have added references at the end of each sentence (Lines 341-342).

An earlier study reported increased levels of CGRP in motoneurons of cat and mouse sciatic nerves 2 to 5 days after axotomy surgery [112].

Lines 358-364: please add multiple references at the end of each of these sentences.

Response: We have added references at the end of each sentence (Lines 354-362).

In vitro and in vivo studies have demonstrated the presence of GAP-43 immunoreactivity in Schwann cell precursors and mature, non-myelin-forming Schwann cells, but not in mature myelin-forming Schwann cells [117]. After denervation, post-axonal GAP-43 expression was shown to be upregulated in almost all Schwann cells in the distal stump [118]. GAP-43 is also expressed by specific CNS glial cells in tissue culture and in vivo, indicating that its expression is not restricted to neurons [119]. Upregulation of GAP-43 in the facial nucleus following compression injury was shown to promote axon growth and regeneration of damaged nerves [120]. GAP-43 mRNA and protein are upregulated early after axonal injury and gradually decrease as the nerve recovers [121].

Lines 393-407: please add multiple references at the end of each of these sentences.

Response: We have added references at the end of each sentence (Lines 390-403).

After nerve transection, reactive Schwann cells in the remaining distal nerve produce a range of trophic factors, including BDNF, nerve growth factor (NGF), and neurotrophin-4 (NT-4) [129]. After facial nerve transection, BDNF mRNA is increased in brainstem nuclei and the thalamus; in addition to BDNF mRNA, BDNF protein is increased in the facial nucleus after axonal transection of the facial nerve, where it may contribute to the survival of motor neurons [130].

Mature BDNF preferentially binds to TrkB, a high-affinity receptor for BDNF, to elicit growth-promoting signals [131]. TrkB is a prototypical tyrosine kinase that dimerizes and autophosphorylates upon ligand binding [132]. Activation of TrkB by BDNF leads to the activation of several intracellular signaling pathways, including the MAPK/ERK and PI3K/Akt pathways, which promote neuronal survival, differentiation, and growth [131]. Neurotrophin signaling through TrkB is involved in protecting against axotomy-induced apoptosis and death of CNS neurons; consistent with this, the survival rate of axotomized hippocampal and motor neurons is low in TrkB‑/‑ mice [133].

  1. Biomolecules decreased in the CNS after peripheral facial nerve injury

Lines 423-425:  please add multiple references at the end of this sentence.

Response: We have added references at the end of each sentence (Lines 422-428).

ChAT can serve as a marker for nerve regeneration after facial nerve damage since the gradual increase in the number of active ChAT-expressing neurons leads to the recovery of facial motor function [131].

Lines 427-428: please add multiple references at the end of this sentence.

Response: We have added references at the end of each sentence (Lines 427-428).

One week after facial nerve axotomy, ChAT immunoreactivity in the facial nucleus is significantly reduced, and after 2 months, ChAT expression increases in many motor neurons [114] (Table 2).

Table 2: please correct in the same way with Table 1. See my proposal above.

  Response: Following the reviewer's advice, we have defined the abbreviations in Tables 1-4 at the bottom of the table. We have also added an abbreviation list before the References section.

Lines 445-446: please add multiple references at the end of this sentence.

Response: We have added references at the end of each sentence (Lines 453-456).

Glycogen is an important energy reserve that can be rapidly mobilized to meet increased energy demands in response to cellular stress [148]. In the nervous system, glycogen is predominantly found in astrocytes, which store and release glycogen-derived glucose to support neuronal activity and maintain energy homeostasis [149].

Lines 464-480: please add multiple references at the end of each of these sentences.

Response: We have added references at the end of each sentence (Lines 463-481).

The reduction in GS protein in severed motor neurons blocks energy-intensive glycogen synthesis, which is used for energy conservation and survival [152]. In sciatic nerve injury, glycogen phosphorylase (GP), which catalyzes the breakdown of glycogen to glucose, is increased in damaged motor neurons and can be used to generate molecules essential for survival [153]. GS plays an important role in glycogen metabolism and energy homeostasis in the nervous system, and its regulation is tightly linked to neuronal function and survival [154].

4.4. m2 muscarinic acetylcholine receptor and nicotinic acetylcholine

receptor

Acetylcholine regulates neuronal differentiation during early development, and both muscarinic and nicotinic acetylcholine receptors regulate a variety of physiological responses, including apoptosis, cell proliferation, and neuronal differentiation [155,156]. Muscarinic receptors are GPCRs that mediate the response to acetylcholine released by parasympathetic nerves [157]. The m2 muscarinic acetylcholine receptor (m2MAchR) is essential for the regulation of various physiological functions, including cardiovascular function and smooth muscle contraction, through activation of G protein-coupled endogenous potassium channels [158,159]. GPCRs bind ligands outside the cell and selectively bind and activate specific G proteins to trigger events inside the cell [160].

Lines 491-493: please add multiple references at the end of this sentence.

Response: We have added references at the end of each sentence (Lines 491-493).

Postganglionic dissection by cervical ganglion axotomy has been shown to reduce mRNA transcripts for α3, α5, α7, and β4 nAChR subunits, and protein expression of α7 and β4 subunits [164].

Lines 497-499: please add multiple references at the end of this sentence.

Response: We have added references at the end of each sentence (Lines 500-502).

Oligodendrocyte myelin glycoprotein (OMgp), which is expressed in neurons and oligodendrocytes in the CNS, is a membrane-anchored protein tethered by a glycosylphosphatidylinositol moiety [167,168].

Lines 510-520: please add multiple references at the end of each of these sentences.

Response: We have added references at the end of each sentence (Lines 507-523).

This indicates that the decrease in OMgp expression is attributable to its downregulation in facial motor neurons rather than oligodendrocytes, and that the change in neuronal OMgp expression is likely involved in the reconnection of neural circuits between axonal facial neurons and upper motor neurons after amputation [169].

4.6. GABAA and GABAB receptors

Facial neurons receive strong GABAergic innervation and are endowed with numerous GABAA and GABAB receptors. GABAA receptors (GABAARs) are ligand-gated chloride channel complexes formed from receptor subunits classified into four families-α (1-6), β (1-3), γ (1-5), and δ-according to sequence similarity [170]. In the mammalian brain, they primarily provide rapid inhibition, mainly as αβγ, αβδ or ρ heteromeric combinations of pentamers [171]. Excitatory neurotransmission is increased after facial nerve axotomy in association with a decrease in GABAA expression in the cell body of motor neurons in facial nuclei, reflecting downregulation of α1, β2, and γ2 mRNAs [146]. This results in changes in the properties, or a reduction in the synaptic transmission, of GABAergic inputs. Changes in Schwann cell-axon connections provide a signaling mechanism to the cell body to downregulate the α1 subunit [172].

Lines 530-541: please add multiple references at the end of each of these sentences.

Response: We have added references at the end of each sentence (Lines 529-545).

GABAB receptors belong to the superfamily of seven transmembrane domain-containing GPCRs and are linked to Ca2+ and K+ channels by G protein and second messenger transduction pathways. After facial nerve axotomy, GABAB receptor levels were reported to increase in the facial nucleus [174]. However, it has also been reported that the abundance of mRNA for GABA (B1B) and GABA (B2) subunits is reduced in motoneurons after facial nerve axotomy, in association with a corresponding change in protein levels of the GABA (B2) subunit, but not the GABA (B1B) subunit [146].

4.7. α-amino-3-hydroxy-5-methylisoxazole-4-propionic acid receptor

and N-methyl-D-as partate receptor

Facial motor neurons receive inputs from various sources, including premotor neurons from the trigeminal nucleus and glutamate nerve endings from the sublingual nucleus and reticular body [175]. These inputs are mediated by glutamate receptors, including α-amino-3-hydroxy-5-methylisoxazole-4-propionic acid (AMPA) receptors and N-methyl-D-aspartate (NMDA) receptors [176]. AMPA and NMDA receptors are ionotropic glutamate receptors that play important roles in synaptic transmission and plasticity in the nervous system [177].

Lines 549-563: please add multiple references at the end of each of these sentences.

Response: We have added references at the end of each sentence (Lines 549-563).

On the other hand, NMDA receptors, composed of NR1 and NR2 subunits, are involved in synaptic development and plasticity [178]. The downregulation of NR1 subunit in facial motor neurons after nerve injury and the detection of NR2A and NR2B subunits in their cell body may have an impact on the plasticity and function of these neurons, which are crucial for motor control and recovery following injury [179].

4.8. Vesicular glutamate transporter

Glutamate, the main excitatory neurotransmitter in the brain, is transported into synaptic vesicles by three types of vesicular glutamate transporters (VGLUTs): VGLUT1, VGLUT2, and VGLUT3 [180]. These transporters are responsible for packaging glutamate into vesicles in the presynaptic terminal; from here it can then be released into the synaptic cleft and bind to glutamate receptors on the postsynaptic membrane, leading to excitatory synaptic transmission [181]. VGLUT2 is expressed in canonical glutamatergic neurons. VGLUT1 and VGLUT2 expression were reported to be reduced in lumbar dorsal root ganglia of rats following sciatic nerve axotomy [182], which was also reported to reduce VGLUT1 immunoreactivity in the spinal cord [183].

Lines 567-569: please add multiple references at the end of this sentence.

Response: We have added references at the end of each sentence (Lines 564-567).

After facial nerve transection, VGLUT2 is reduced in the facial nucleus, as evidenced by a large decrease in VGLUT2 staining [147]; this reduction, which likely serves to protect facial motor neurons from excitotoxic effects, is associated with upregulation of glutamate transporters in activated microglia in facial nuclei [184].

Line 572: please add multiple references at the end of this sentence.

Response: We have added references at the end of each sentence (Lines 571-574).

Post-synaptic density-95 (PSD-95), a scaffolding protein localized to the post-synaptic density, plays a key role in signal transduction, synaptic plasticity and synaptogenesis, and facilitates NO synthesis by clustering NMDARs on synaptic membranes and binding to neuronal NOS (nNOS) [185].

Lines 582-590: please add multiple references at the end of each of these sentences.

Response: We have added references at the end of each sentence (Lines 586-592).

5.1 Autophagy

Autophagy, a cytoprotective process commonly found in eukaryotes, plays a critical role in maintaining cellular and tissue homeostasis by removing damaged organelles, pathological proteins, and dysfunctional macromolecules from within cells [187]. Autophagy regulates various physiological and pathological processes through the lysosomal degradation pathway, including nerve regeneration, myelin development, myelin degradation, and neurodegeneration [188]. Autophagy is an initial process activated after facial nerve injury that serves to repair the damage [189].

  1. Central facial nerve biomolecules and processes involved in peripheral facial nerve damage

Lines 601-602: please add multiple references at the end of this sentence.

Response: We have added references at the end of each sentence (Lines 599-602).

Treatment of facial nerve injury with basic fibroblast growth factor (bFGF) restored the morphology and function of the damaged facial nerve by promoting autophagy and inhibiting apoptosis through activation of the P21-activated kinase 1 (PAK1) signaling pathway [193] (Table 3).

Table 3: please correct in the same way with Table 1. See my proposal above.

  Response: Following the reviewer's advice, we have defined the abbreviations in Tables 1-4 at the bottom of the table. We have also added an abbreviation list before the References section.

Lines 624-637: please add multiple references at the end of each of these sentences.

Response: We have added references at the end of each sentence (Lines 628-640).

5.3. Interleukin-10

Interleukin-10 (IL-10) is a cytokine that plays a crucial role in regulating inflammation and immune responses [198]. In the CNS, IL-10 is upregulated in a number of pathological contexts, such as cerebral artery occlusion, excitotoxicity, and traumatic brain injury [199]. Astrocytes and microglia are potential sources of IL-10 production, and IL-10 receptors are expressed on microglia, astrocytes, oligodendrocytes, and neurons [200]. The inflammation- and immune-regulatory functions of IL-10 are critical for the survival of facial motor neurons following facial nerve injury [201]. Consistent with this, mice with selective knockout of IL-10 exhibit decreased facial nerve survival compared to wild-type mice [202]. IL-10 produced by neurons and astrocytes plays an important role in maintaining neuronal cell homeostasis and providing neuroprotective nutrition after axotomy [203]. After facial nerve axotomy, IL-10 plays a vital role in maintaining an anti-inflammatory environment in the CNS and is produced by several other cells, including T helper 2 (Th2) cells, to exert a direct anti-apoptotic effect on neurons [204].

Lines 648-657: please add multiple references at the end of each of these sentences.

Response: We have added references at the end of each sentence (Lines 647-660).

5.4. Calcium

Calcium is involved in several signaling pathways that regulate cellular homeostasis [205]. Calcium signaling is critical to the glial environment of neurons and plays an important role in neuron survival and regeneration after axonal injury [206]. The expansion of excitotoxic glutamate resulting from stroke, epilepsy, or traumatic brain injury leads to an elevation in intracellular calcium ions that, if excessive, can trigger processes leading to neuronal cell death and necrosis [207]. Elevated cytosolic calcium levels have been found in axons and soma of mechanoreceptor neurons within minutes after axotomy, indicating the importance of calcium signaling in the response to axonal injury [208]. Disrupting calcium influx may be useful in counteracting injury-induced neuronal cell death [209]. Nimodipine has been shown to have neuroprotective effects after various lesions in the CNS, including in the facial motor nucleus after intracranial transection of the facial nerve [210]. Thus, nimodipine and other calcium antagonists may be useful in protecting neurons from injury-induced cell death and promoting regeneration after axonal injury [211].

Line 661: please add multiple references at the end of this sentence.

Response: We have added references at the end of each sentence (Lines 664-665).

Cortical plasticity, which refers to the cortex’s ability to adapt to a changing environment and new information, occurs in brain and peripheral nerve lesions [212].

Lines 668-671: please add multiple references at the end of each of these sentences.

Response: We have added references at the end of each sentence (Lines 668-675).

An analysis of changes in facial nerve morphology 4 weeks after facial nerve transection revealed changes in electrophysiological properties and firing frequency adaptation in two types of putative facial nucleus motoneurons [213]. Specifically, it was shown that firing rates are reduced and firing patterns are altered in motoneurons in the context of facial nerve palsy and neurotmesis due to facial nerve neuropathy. This suggests that peripheral facial nerve injury can cause changes in the electrophysiological properties and firing frequency adaptation of motoneurons in the facial nucleus [10].

Lines 676-687: please add multiple references at the end of each of these sentences.

Response: We have added references at the end of each sentence (Lines 678-681).

After recovery from facial nerve palsy, neurons innervating the zygomatic muscle, identified using retrograde staining with horseradish peroxidase (HRP), were located in the facial nucleus; labeled neurons were also found in the facial nucleus region following injection of HRP into the orbicularis, zygomatic, and orbicularis oris muscles [214,215].

Table 4: please correct in the same way with Table 1. See my proposal above.

   Response: Following the reviewer's advice, we have defined the abbreviations in Tables 1-4 at the bottom of the table. We have also added an abbreviation list before the References section.

Discussion

Lines 705-752: please add multiple references at the end of each of these sentences.

Response: We have added references at the end of each sentence (Lines 700-764).

One critical process for maintaining cellular and tissue homeostasis is autophagy, which removes damaged organelles, pathological proteins, and dysfunctional macromolecules from within cells. Macrophages are crucial players in nerve injury and regeneration. After nerve injury, macrophages trigger an inflammatory response and help to clear and regenerate damaged tissue [217]. Moreover, the coordinated actions of M1 and M2 subtypes can establish a favorable microenvironment for the release of cytokines that aid in the repair of damaged tissue [218]. As a result, macrophages have a critical role in both nerve damage and regeneration. After facial nerve injury, activation of autophagy is beneficial for reducing scarring and promoting myelination, which in turn help to facilitate regeneration and motor recovery[190]. ROS levels also increase significantly after peripheral nerve injury, leading to cellular damage through oxidative stress [219]. Thus, inhibiting ROS production could mitigate oxidative damage induced by facial nerve injury. Moreover, IL-10 plays a significant role in regulating inflammatory and immune responses following facial nerve injury [204]. Inhibition of IL-10 reduces facial nerve survival, whereas overproduction of IL-10 protects against facial nerve injury by preventing neuronal cell death.

After peripheral facial nerve injury, activated PS and SHARPIN act on neurons and glial cells in the CNS to support cell survival [49,220]. Additionally, increased release of NO promotes circulation at the damaged site by inducing local vasodilation [78]. iNOS-mediated NO production also has a beneficial effect on the host’s defense mechanisms against bacteria, but excessive increases in NO can lead to cytotoxicity and cell death. As first responders to injury, VIP and PS help resolve acute inflammatory processes and inhibit the induction of TNFα and IL-6 in CNS microglia, which reduces neuronal loss around the lesion site [85]. FGF-2 plays an important role in tissue development and damage repair, increasing the number of GFAP-positive astrocytes in the facial nucleus after facial nerve injury and enhancing their expression of FGF 2 [93,98]. FGF-2 promotes the survival of injured neurons and preserves synaptic plasticity and neurotransmitter release, which are crucial for neuroprotection [89]. Shh and Smo, which plays regulatory role in transmitting information necessary for cell differentiation, are upregulated in facial motor neurons, where their increased activity serves to restore the synaptic transmission necessary for nerve repair after facial neurectomy [105]. CGRP and GAP-43, derived from motor neuron cell bodies, are involved in nerve regeneration repair mechanisms; their activity promotes the regeneration of damaged nerves by positively affecting axon regeneration and growth [115,119]. Overexpression of GDNF protects motor neurons from apoptosis and promotes cell survival [125]. Activation of its GDNFR α/c-ret receptor complex contributes to facial nerve regeneration in motor neurons in the spinal cord and brainstem nuclei, thereby promoting axonal regeneration [124]. BDNF and TrkB are increased in the facial nucleus after facial nerve injury and help maintain CNS homeostasis by preventing neuronal cell death [130]. These factors and signaling pathways all work together to promote tissue repair and recovery after facial nerve injury.

After facial nerve axotomy, cholinergic neurotransmitters in the facial nucleus of the CNS are down-regulated, and the number of ChAT neurons the primary motor neurons involved in acetylcholine synthesis is reduced [137]. However, the expression of ChAT increases with the recovery of facial motor function and can be used as a functional marker of nerve regeneration. Furthermore, a decrease in m2MAchR indicates a decrease in motor neurons, and a decrease in nAChR stimulates the production of pro-inflammatory molecules [163].

In addition, the excitatory neurotransmitter, glutamate, increases after facial nerve transection, as does the expression of GABAA, KCC2 and gephyrin, indicating a decrease in inhibitory neurotransmission [145,146]. However, GABAB receptor levels are increased, notably in microglial cells of the facial nucleus [174]. Following facial nerve transection, GS levels are reduced in the facial nucleus, thereby disrupting the synthesis of glycogen, an essential molecule used for energy conservation and survival [137]. OMgp is expressed in neurons and oligodendrocytes, and its expression in oligodendrocytes inhibits axon regeneration after central nervous system injury. After nerve injury, OMgp expression was found to decrease on day 14 and return to normal levels after day 28 [169]. This expression pattern was not observed in oligodendrocytes, but was observed in neurons. This suggests that the decrease in OMgp after facial nerve injury is not related to changes in the expression of oligodendrocytes, but rather to changes in the expression of neurons, and implies that changes in OMgp expression in neurons are involved in the reconnection of the neural circuit between axonal facial neurons and upper motor neurons after injury[169]. The reduction in AMPA and NMDA receptors in the axotomized facial nucleus may result in excitotoxicity of facial motor neurons owing to increases in intracellular calcium concentration [147]. An excessive increase in calcium causes neuronal cell death and necrosis, whereas reducing calcium influx after facial nerve injury may increase neuronal survival.

Round 2

Reviewer 2 Report

The authors have responded appropriately to all questions I raised. I suggest the editor may consider accepting this manuscript to be published in its current state.

Reviewer 3 Report

The pictures are informative and ok. The ms decision: accept